# Collaborative Causal Discovery
# with Atomic Interventions

**Raghavendra Addanki**
University of Massachusetts, Amherst
raddanki@cs.umass.edu

**Shiva Prasad Kasiviswanathan**
Amazon
kasivisw@gmail.com

## Abstract

We introduce a new Collaborative Causal Discovery problem, through which we model a common scenario in which we have multiple independent entities each with their own causal graph, and the goal is to simultaneously learn all these causal graphs. We study this problem without the causal sufficiency assumption, using Maximal Ancestral Graphs (MAG) to model the causal graphs, and assuming that we have the ability to actively perform independent single vertex (or atomic) interventions on the entities. If the $M$ underlying (unknown) causal graphs of the entities satisfy a natural notion of clustering, we give algorithms that leverage this property, and recovers all the causal graphs using roughly logarithmic in $M$ number of atomic interventions per entity. These are significantly fewer than $n$ atomic interventions per entity required to learn each causal graph separately, where $n$ is the number of observable nodes in the causal graph. We complement our results with a lower bound and discuss various extensions of our collaborative setting.

## 1 Introduction

In this paper, we introduce a new model for *causal discovery*, the problem of learning all the causal relations between variables in a system. Under certain assumptions, using just observational data, some ancestral relations as well as certain causal edges can be learned, however, many observationally equivalent structures cannot be distinguished [Zhang, 2008a]. Given this issue, there has been a growing interest in learning causal structures using the notion of an *intervention* described in the Structural Causal Models (SCM) framework introduced by Pearl [2009].

As interventions are expensive (require carefully controlled experiments) and performing multiple interventions is time-consuming, an important goal in causal discovery is to design algorithms that utilize simple (preferably, single variable) and fewer interventions [Shanmugam et al., 2015]. However, when there are latents or unobserved variables in the system, in the worst-case, it is not possible to learn the exact causal DAG without intervening on every variable at least once. Furthermore, multivariable interventions are needed in presence of latents [Addanki et al., 2020].

Figure 1: Examples of $M$ causal graphs constructed from Lung Cancer dataset [Lauritzen and Spiegelhalter, 1988]. Here, the causal graphs differ only in the presence of latents (nodes with dotted square box), but they could differ elsewhere too.

On the other hand, in a variety of applications, there is no one true causal structure, different entities participating in the application might have different causal structures [Gates and Molenaar, 2012, Ramsey et al., 2011, Joffe et al., 2012]. For example, see figure 1. In these scenarios, generating a single causal graph by pooling data from these different entities might lead to flawed conclusions [Saeed et al., 2020]. Allowing for interventions, we propose a new model for tackling this problem, referred here as *Collaborative Causal Discovery*, which in its simplest form states that: given a collection of entities, each associated with an individual unknown causal graph and generating their own independent data samples, learn all the causal graphs while minimizing the number of *atomic* (single variable) interventions for every entity. An underlying assumption is that each entity on its own generates enough samples in both the observational and interventional settings so that conditional independence tests can be carried out *accurately* on each entity separately. To motivate this model of collaborative causal discovery, let us consider two different scenarios.

**(a)** Consider a health organization interested in controlling incidence of a particular disease. The organization has a set of $M$ individuals (entities) whose data it monitors and can advise interventions on. Each individual is an independent entity that generates its own set of separate data samples[1]. In a realistic scenario, it is highly unlikely that all the $M$ individuals share the same causal graph (e.g., see Figures 3a and 3b from Joffe et al. [2012] in Appendix A). It would be beneficial for the organization to collaboratively learn all the causal graphs together. The challenge is, *a priori* the organization does not know the set of possible causal graphs or which individual is associated with which graph from this set.

**(b)** An alternate setting is where, we have $M$ companies (entities) wanting to work together to improve their production process. Each company generates their own data (e.g., from their machines) which they can observe and intervene on [Nguyen et al., 2016]. Again if we take the $M$ causal graphs (one associated with each company) it is quite natural to expect some variation in their structure, more so because we do not assume *causal sufficiency* (i.e., we allow for latents). Since interventions might need expensive and careful experimental organization, each company would like to reduce their share of interventions.

The collaborative aspect of learning can be utilized if we assume that there is some underlying (unknown) clustering/grouping of the causal graphs on the entities.

**Our Contributions**. We formally introduce the collaborative causal discovery problem in Section 2. We assume that we have a collection of $M$ entities that can be partitioned into $k$ clusters such that any pair of entities belonging to two different clusters are separated by large distance (see Definition 2.1) in the causal graphs. Due to presence of latents variables, we use a family of mixed graphs known as *maximal ancestral graphs* (MAGs) to model the graphs on observed variables. Each entity is associated with a MAG.

In this paper, we focus on designing algorithms that have *worst-case* guarantees on the number of atomic interventions needed to recover (or approximately recover) the MAG of each entity. We assume that there are $M$ MAGs one for each entity over the same set of $n$ nodes. Learning a MAG with atomic interventions, in worst case requires $n$ interventions (see Proposition 3.2). We show that this bound can be substantially reduced if the $M$ MAGs satisfy the property that every pair of MAGs from different clusters have *at least* $\alpha n$ nodes whose direct causal relationships are different. We further assume that entities belonging to same cluster have similar MAGs in that every pair of them have *at most* $\beta n$ ($\beta < \alpha$) nodes whose direct causal relationships are different. We refer to this clustering of entities as $(\alpha, \beta)$-clustering (Definition 2.2). A special but important case is when $\beta = 0$, in which case all the entities belonging to the same cluster have the same causal MAG (referred to as $\alpha$-clustering, Definition 2.3). An important point to notice is that while we assume there is an underlying clustering on the entities, it is *learnt* by our algorithms. Similar assumptions are common for recovering the underlying clusters, in many areas, for e.g., crowd-sourcing applications [Ashtiani et al., 2016, Awasthi et al., 2012].

We first start with the observation that under $(\alpha, \beta)$-clustering, even entities belonging to the same cluster could have a different MAG, which makes exact recovery hard without making a significant number of interventions per entity. We present an algorithm that using at most $O(\Delta \log(M/\delta)/(\alpha - \beta)^2)$ many interventions per entity, with probability at least $1 - \delta$ (over only the randomness of the algorithm), can provably recover an *approximate* MAG for each entity. The approximation is such

---

[1] As is common in causal discovery, for the underlying conditional independence tests, the data is assumed to be i.i.d. samples from the interventional/observational distributions.

that for each entity we generate a MAG that is at most $\beta n$ node-distance from the true MAG of that entity (see Section 3). Here, $\Delta$ is the maximum undirected degree of the causal MAGs. Our idea is to first recover the underlying clustering of entities by using a randomized set of interventions. Then, we distribute the interventions across the entities in each cluster, thereby, ensuring that the number of interventions per entity is small. By carefully combining the results learnt from these interventions we construct the approximate MAGs. For the number of interventions, the linear dependence on $\Delta$ is not uncommon for learning causal graphs [Kocaoglu et al., 2017]. Moreover, most real-world causal bayesian networks are known to have small maximum degrees (see section 5).

Under the slightly more restrictive $\alpha$-clustering assumption, we present algorithms that can *exactly* recover all the MAGs using at most $\min\left\{O(\Delta \log(M/\delta)/\alpha), O(\log(M/\delta)/\alpha + k^2)\right\}$ many interventions per entity (see Section 4). Again, randomization plays an important role in our approach.

Complementing these upper bounds, we give a lower bound using Yao's minimax principle [Yao, 1977] that shows for any (randomized or deterministic) algorithm $\Omega(1/\alpha)$ interventions per entity is required for this causal discovery problem. This implies the $1/\alpha$ dependence in our upper bound in the $\alpha$-clustering case is optimal.

Finally, a note about parameters. The $(\alpha, \beta)$-clustering is universal, in the sense that *any* collection of MAGs will satisfy the $(\alpha, \beta)$-clustering property for some value of $\alpha, \beta$ (with $\alpha > \beta$). Ideally, we would like in our problem instance, $\alpha$ to be close to $1$ and $\beta$ to be close to $0$. In most real-world applications, we would also expect $k$ to be relatively small and $M \gg n, k$.

In Section 5, we show experiments on data generated from both real and synthetic networks with added latents and demonstrate the efficacy of our algorithms for learning the underlying clustering and the MAGs.

**Related Work.** A number of algorithms, working under various assumptions, for learning causal graph (or a causal DAG) using interventions have been proposed in the literature, e.g., [Eberhardt, 2007, Hyttinen et al., 2013, Hu et al., 2014, Shanmugam et al., 2015, Kocaoglu et al., 2017, Ghassami et al., 2018, Lindgren et al., 2018, Acharya et al., 2018, Bello and Honorio, 2018, Kocaoglu et al., 2019, Greenewald et al., 2019, Jaber et al., 2020, Addanki et al., 2020, 2021, Tadepalli and Russell, 2021]. Saeed et al. [2020] consider a model where the observational data is from a mixture of causal DAGs, and outline ideas that recover a *union graph* (up to Markov equivalence) of these DAGs, without any interventions. Our setting is not directly comparable to theirs, as we have entities generating data and doing conditional independence tests independently (no pooling of data from entities), but show stronger guarantees for recovering causal graphs, assuming atomic interventions.

## 2 Our Model and Problem Statement

In this section, we introduce the collaborative causal discovery problem. We start with some notations.

**Notation.** Following the SCM framework [Pearl, 2009], we represent the set of random variables of interest by $V \cup L$ where $V$ represents the set of endogenous (observed) variables that can be measured and $L$ represents the set of exogenous (latent) variables that cannot be measured. We do not deal with selection bias in this paper. Let $|V| = n$.

We assume that the causal Markov condition and faithfulness holds for both the observational and interventional distributions [Hauser and Bühlmann, 2012]. We use conditional independence (CI) tests of the form $u \perp\!\!\!\perp v \mid Z$ or $u \perp\!\!\!\perp v \mid \mathrm{do}(w), Z$, for some $u, v, w \in V$ and $Z \subseteq V$ (See Appendix A for more details).

Throughout this paper, unless otherwise specified, a path between two nodes is an undirected path. A path of only directed edges is called a directed path. $u$ is called an ancestor of $v$ and $v$ a descendant of $u$ if $u = v$ or there is a directed path from $u$ to $v$. A directed cycle occurs in $G$ when $u \to v$ is in $G$ and $v$ is an ancestor of $u$.

**Our Model.** We assume that we have access to $M$ entities labeled $1, \ldots, M$, each of which can independently generate their own observational and interventional data. Each entity $i$ has an associated causal DAG $\mathcal{D}_i$ over $V \cup L_i$, where $L_i$ represents the latent variables of entity $i$. In modeling the problem of causal discovery, complications arise in at least two ways:

**(i) Latents**. We allow some variables (called latents) in the causal DAG to be unobservable. As regular DAGs are not sufficient to represent the observed distribution when there are latents, we use *ancestral graphical models* that have been proposed as an elegant and useful surrogate for DAG models with latent variables [Richardson and Spirtes, 2002].

A mixed graph containing directed ($\leftarrow$) and bidirected ($\leftrightarrow$) edges is said to be *ancestral* if it has no directed cycles, and whenever there is a bidirected edge $u \leftrightarrow v$, then there is no directed path from $u$ to $v$ or from $v$ to $u$. An ancestral graph on $V$ (observables) is said to be *maximal*, if, for every pair of nonadjacent vertices $u, v$, there exists a set $Z \subset V$ with $u, v \notin Z$ such that $u$ and $v$ are $m$-separated (similar to $d$-separation, see Definition A.2) conditioned on $Z$. Every DAG with latents (and selection variables) can be transformed into a unique maximal ancestral graph (MAG) over the observed variables [Richardson and Spirtes, 2002].

**(ii) Uniqueness**. Secondly, with just observational data, if the MAGs $\mathcal{M}_1, \ldots, \mathcal{M}_M$ are Markov equivalent, then, without additional strong assumptions they cannot be distinguished, even if they are all structurally different. To overcome the problem of being not identifiable within an equivalence class, we allow for interventions on observed variables. In particular, we focus on atomic interventions in this paper, which are the simplest and most commonly used intervention type, modeled through the do-operator [Pearl, 1995]. As it turns out, Maximal Ancestral Graphs (MAGs) are uniquely identifiable using atomic interventions.[2]

Our objective will be to minimize these interventions. In particular, since each of these entities independently generate their own data, so we aim to reduce the number of interventions needed per entity. In causal discovery, minimizing the number of interventions while ensuring that they are of small size is an active research area [Pearl, 1995, Shanmugam et al., 2015, Ghassami et al., 2018, 2019].

Given the $M$ entities, let $\mathcal{M}_i$ denote the MAG associated with entity $i$ (the MAG constructed from the DAG $\mathcal{D}_i$). Our goal is to collaboratively learn all these MAGs $\mathcal{M}_1, \ldots, \mathcal{M}_M$ while minimizing the maximum number of interventions per entity.

To facilitate this learning, we make a natural underlying clustering assumption that partitions the entities based on their respective MAGs such that: (i) any two entities belonging to the same cluster have MAGs that are "close" to each other, (ii) any two entities belonging to different clusters have MAGs that are "far" apart. Before stating this assumption formally, we need some definitions.

For MAG $\mathcal{M}_i = (V, E_i)$, we denote the children (through outgoing edges), parent (through incoming edges), and spouse (through bidirected edges) of a node $u \in V$ as

$$\text{ch}_i(u) = \{v \mid u \rightarrow v \in E_i\}, \ \text{pa}_i(u) = \{v \mid u \leftarrow v \in E_i\}, \ \text{sp}_i(u) = \{v \mid u \leftrightarrow v \in E_i\}. \quad (1)$$

Also, define an incidence set for a vertex $u \in V$ which contains an entry $(v, \text{type})$ for every node $v$ adjacent to $u$ as

$$N_i(u) = \left\{ \begin{array}{ll} (v, \text{tail}) & \text{if } u \rightarrow v \in E_i \\ (v, \text{head}) & \text{if } u \leftarrow v \in E_i \\ (v, \text{bidirected}) & \text{if } u \leftrightarrow v \in E_i \end{array} \right\}. \quad (2)$$

Note that $|N_i(u)|$ is the undirected degree of $u$ in $\mathcal{M}_i$. We now define a distance measure between MAGs that captures structural similarity between them.

**Definition 2.1.** *Given two MAGs $\mathcal{M}_i = (V, E_i)$ and $\mathcal{M}_j = (V, E_j)$, define the node-difference as the set: $\text{diff}(\mathcal{M}_i, \mathcal{M}_j) = \{u \in V \mid N_i(u) \neq N_j(u)\}$, and the node-distance as the cardinality of this set: $d(\mathcal{M}_i, \mathcal{M}_j) = |\text{diff}(\mathcal{M}_i, \mathcal{M}_j)| = |\{u \in V \mid N_i(u) \neq N_j(u)\}|$.*

Intuitively, the node distance captures the number of nodes whose incidence relationships differ. It is easy to observe that the node distance is a distance metric, and captures a strong structural similarity between the graphs. Two graphs $\mathcal{M}_i, \mathcal{M}_j$ are identical iff $d(\mathcal{M}_i, \mathcal{M}_j) = 0$. For e.g., in Figure 2, we have two MAGs that satisfy $d(\mathcal{M}_{12}, \mathcal{M}_{13}) = 2$ as $\text{diff}(\mathcal{M}_{12}, \mathcal{M}_{13}) = \{x, z\}$, where $d(\mathcal{M}_{12}, \mathcal{M}_{21}) = 3$ as $\text{diff}(\mathcal{M}_{12}, \mathcal{M}_{21}) = \{x, y, z\}$. We are now ready to define a simple clustering property on MAGs.

---

[2]However, in the presence of latents, even with power of atomic interventions, the structure of a causal DAG is not uniquely identifiable. (see, e.g., In Figure 4 in Appendix A). Similarly, we can show that using single vertex interventions, we also cannot exactly recover a wider class of acyclic graphs like ADMGs (Acyclic Directed Mixed Graphs).

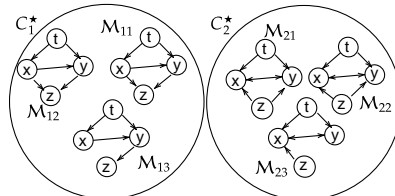

Figure 2: MAGs with $(\alpha = 0.75, \beta = 0.5)$-clustering. Every pair of graphs in $C_1^\star$ and $C_2^\star$ differ in at least $3(= 0.75 \times 4)$ nodes, while pairs of graphs within clusters differ by at most $2(= 0.5 \times 4)$ nodes.

| **Algorithm 1** IDENTIFY-OUTNBR $(\mathcal{U}_i, u)$ | **Algorithm 2** IDENTIFY-BIDIRECTED $(\mathcal{U}_i, u)$ |
|---|---|
| 1: **Input:** node $u \in V$, PAG $\mathcal{U}_i$ of entity $i$ | 1: **Input:** node $u \in V$, PAG $\mathcal{U}_i$ of entity $i$ |
| 2: **Output:** $\mathrm{ch}_i(u)$ | 2: **Output:** $\mathrm{sp}_i(u)$ |
| 3: $\mathrm{ch}_i(u) = \{v \mid u \rightarrow v \in \mathcal{U}_i\}$ | 3: $\mathrm{sp}_i(u) = \{v \mid u \leftrightarrow v \in \mathcal{U}_i\}$ |
| 4: **for** $v \in \Gamma_i(u)$ such that $u \circ\!\!-\!\!\circ v$ or $u \circ\!\!\rightarrow v \in \mathcal{U}_i$ **do** | 4: **for** $v \in \Gamma_i(u)$ such that $u \circ\!\!-\!\!\circ v$ or $u \leftarrow\!\!\circ v$ or $u \circ\!\!\rightarrow v \in \mathcal{U}_i$ **do** |
| 5:    **if** $u \not\!\perp\!\!\!\perp v \mid \mathrm{do}(u)$ **then** | 5:    **if** $u \perp\!\!\!\perp v \mid \mathrm{do}(u)$ and $u \perp\!\!\!\perp v \mid \mathrm{do}(v)$ **then** |
| 6:      $\mathrm{ch}_i(u) \leftarrow \mathrm{ch}_i(u) \cup \{v\}$ | 6:      $\mathrm{sp}_i(u) \leftarrow \mathrm{sp}_i(u) \cup \{v\}$ |
| 7:    **end if** | 7:    **end if** |
| 8: **end for** | 8: **end for** |
| 9: Return $\mathrm{ch}_i(u)$ | 9: Return $\mathrm{sp}_i(u)$ |

**Definition 2.2** (($\alpha, \beta$)-clustering). *Let $\mathcal{M}_1, \ldots, \mathcal{M}_M$ be a set of $M$ MAGs. We say that this set of MAGs satisfy the $(\alpha, \beta)$-clustering property, with $\alpha > \beta \geq 0$, if there exists a partitioning of $[M]$ into sets (clusters) $C_1^\star, \ldots, C_k^\star \subset [M]$ (for some $k \in \mathbb{N}$) such that for all $(i, j) \in [M] \times [M]$:*

*(i) if $i$ and $j$ belong to same set (cluster), then $d(\mathcal{M}_i, \mathcal{M}_j) \leq \beta n$;*

*(ii) if $i$ and $j$ belong to different sets (clusters), then $d(\mathcal{M}_i, \mathcal{M}_j) \geq \alpha n$.*

Under this definition, all the $M$ MAGs could be different. See, e.g., Figure 2. With right setting of $\alpha > \beta$ we can capture any set of possible $M$ MAGs. Therefore, an algorithm such as FCI [Spirtes et al., 2000], that constructs PAGs might not be able to recover the clusters, as all the PAGs could be different, and the node-distance between PAGs does not correlate well with the node-distance between corresponding MAGs (e.g., see Figure 5 in Appendix A). We use the PAGs generated by FCI as a starting point for all our algorithms and further refine them. We assume that PAGs generated are correct (see Appendix B.1 for additional details). With this discussion, we introduce our collaborative causal graph learning problem as follows:

> **Assumption:** MAGs $\mathcal{M}_1, \ldots, \mathcal{M}_M$ (associated with entities $1, \ldots, M$ respectively) satisfying the $(\alpha, \beta)$-clustering property
> **Access to each entity:** Through conditional independence (CI) tests on observational and interventional distributions. Each entity generates their own (independent) data samples.
> **Goal:** Learn $\mathcal{M}_1, \ldots, \mathcal{M}_M$ while minimizing the max. number of interventions per entity.

An interesting case of the Definition 2.2 is when $\beta = 0$.

**Definition 2.3** ($\alpha$-clustering). *We say a set of MAGs $\mathcal{M}_1, \ldots, \mathcal{M}_M$ satisfy the $\alpha$-clustering property, if and only if they satisfy $(\alpha, 0)$-clustering property.*

Note that $\alpha$-clustering is a natural property, wherein each cluster is associated with a single unique MAG, and all entities in the cluster have the same MAGs, and same conditional independences.

## 3 Causal Discovery under $(\alpha, \beta)$-Clustering Property

In this section, we present our main algorithm for collaboratively learning causal MAGs under the $(\alpha, \beta)$-clustering property. Missing details from this section are presented in Appendix C.

**Definition 3.1** (Partial Ancestral Graph (PAG)). *Let $[\mathcal{M}_i]$ denote the Markov equivalence class of the MAG $\mathcal{M}_i$ and represented by the Partial Ancestral Graph (or PAG) $\mathcal{U}_i = (V, \widehat{E}_i)$. Edges $\widehat{E}_i$ have three kinds of endpoints given by arrowheads ($\leftarrow$), circles ($\circ\!-$) and tails ($-$).*

All our algorithms are randomized, and succeed with high probability over the randomness introduced by the algorithm. The idea behind all our algorithms is to first learn the true clusters $C_1^\star, \ldots, C_k^\star$ using very few interventions. Once the true clusters are recovered, the idea is to distribute the interventions across the entities in each cluster and merge the results learned to recover the MAGs (Section 3.2). For our algorithms, a lower bound for $\alpha$ and upper bound for $\beta$ is sufficient. In practice, a clustering of the PAGs (generated from FCI algorithm) can provide guidance about these bounds on $\alpha, \beta$, or if we have additional knowledge that $\alpha \in [1 - \epsilon, 1]$ and $\beta \in [0, \epsilon]$ for some constant $\epsilon > 0$, then, we can use binary search, that increases our intervention bounds by $\log^2(n\epsilon)/(1 - 2\epsilon)^2$ factor. It is important to note that none of our algorithms require the knowledge of $k$

**Helper Routines.** Let $\Gamma_i(u)$ denote all nodes that are adjacent to $u$ in the PAG $\mathcal{U}_i$, i.e., $\Gamma_i(u) = \{v \mid (u, v) \in \widehat{E}_i\}$. Given the PAG $\mathcal{U}_i$, Algorithm IDENTIFY-OUTNBR identifies all the outgoing neighbors of any node $u$ in $\mathcal{M}_i$. We look at edges of the form $u\circ\!\!-\!\!\circ v$ or $u\circ\!\!\rightarrow v$ in $\mathcal{U}_i$ incident on $u$, and identify if $u \rightarrow v$ using the CI-test $u \perp\!\!\!\perp v \mid \mathrm{do}(u)$. This is based on the observation that any node $v$ that is a descendant of $u$ (including $\mathrm{ch}_i(u)$) satisfies $u \not\perp\!\!\!\perp v \mid \mathrm{do}(u)$. Algorithm IDENTIFY-BIDIRECTED identifies all the bidirected edges incident on $u$. If there is an edge of the form $u\circ\!\!-\!\!\circ v$ or $u\leftarrow\!\!\circ v$ or $u\circ\!\!\rightarrow v$ in the PAG, and $v \notin \mathrm{ch}_i(u)$ and $u \notin \mathrm{ch}_i(v)$, then it must be a bidirected edge.

Using these helper routines, we give an Algorithm RECOVERG (in Appendix B) that recovers any MAG $\mathcal{M}_i$ using $n$ atomic interventions. Complementing this, we show that $n$ interventions are also required. The missing details are presented in Appendix B.

**Proposition 3.2.** *There exists a causal MAG $\mathcal{M}$ such that every adaptive or non-adaptive algorithm requires $\Omega(n)$ many atomic interventions to recover $\mathcal{M}$.*

## 3.1 Recovering the Clusters

From the $(\alpha, \beta)$-clustering definition, we know that a pair of entities belonging to the same cluster have higher structural similarity between their MAGs than a pair of entities across different clusters. Let us start with a simplifying assumption that $\beta = 0$ (i.e., $\alpha$-clustering). So, all the MAGs are separated by a distance of at least $\alpha n$. We make the observation that to identify that two MAGs, say $\mathcal{M}_i$ and $\mathcal{M}_j$ belong to different clusters, it suffices to find a node $u$ from the node-difference set $\mathrm{diff}(\mathcal{M}_i, \mathcal{M}_j)$ and checking their neighbors using Algorithms IDENTIFY-OUTNBR and IDENTIFY-BIDIRECTED. We argue that (see Claim D.3, Appendix D.2), with probability at least $1 - \delta$, we can identify one such node $u \in \mathrm{diff}(\mathcal{M}_i, \mathcal{M}_j)$ by sampling $2\log(M/\delta)/\alpha$ nodes uniformly from $V$ as $|\mathrm{diff}(\mathcal{M}_i, \mathcal{M}_j)| = d(\mathcal{M}_i, \mathcal{M}_j) \geq \alpha n$.[3] However, this approach will not succeed when $\beta \neq 0$ because now we have MAGs in the same cluster that are also separated by non-zero distance.

**Overview of Algorithm $(\alpha, \beta)$-BOUNDEDDEGREE.** We now build upon the above idea, to recover the true clusters $C_1^\star, \ldots, C_k^\star$ when $\beta \neq 0$. As identifying a node $u \in \mathrm{diff}(\mathcal{M}_i, \mathcal{M}_j)$ is not sufficient, we maintain a count of the number of nodes among the sampled set of nodes $S$ that the pair of entities $i, j$ have the same neighbors, i.e., $\mathrm{COUNT}(i, j) = \sum_{u \in S} \mathbf{1}\{N_i(u) = N_j(u)\}$. Based on a carefully chosen threshold value for the $\mathrm{COUNT}(i, j)$, that arises through the analysis of our randomized algorithm, we classify whether a pair of entities belong to the same cluster correctly.

Overall, the idea here is to construct a graph $\mathcal{P}$ on entities (i.e., the node set of $\mathcal{P}$ is $[M]$). We include an edge between two entities $i$ and $j$ if $\mathrm{COUNT}(i, j)$ is above the threshold $(1 - (\alpha + \beta)/2)|S|$. Using Lemma 3.3, we show that this threshold corresponds to the case where if the entities are from same true clusters, then the COUNT value corresponding to the pair is higher than the threshold; and if they are from different clusters it will be smaller, with high probability. This ensures that every entity is connected only to the entities belonging to the same true cluster. We return the connected components in $\mathcal{P}$ as our clusters.

**Theoretical Guarantees.** In Algorithm $(\alpha, \beta)$-BOUNDEDDEGREE, we construct a uniform sample $S$ of size $O(\log(M/\delta)/(\alpha - \beta)^2)$, and identify all the neighbors of $S$ for every entity $i \in [M]$. As we use IDENTIFY-BIDIRECTED to identify all the bi-directed edges, the total number of interventions

---

[3]For theoretical analysis, our intervention targets are randomly chosen, even with the knowledge available from PAGs, because in the worst-case the PAGs might contain no directed edges to help decide which nodes to intervene on. In practice, though if we already know edge orientations from PAG we do not have to relearn them, and a biased sampling based on edges uncertainties in PAGs might be a reasonable approach.

---

**Algorithm 3** $(\alpha, \beta)$-BOUNDEDDEGREE

---

1: **Input:** $\alpha > 0$, $\beta \geq 0$ $(< \alpha)$, confidence parameter $\delta > 0$, PAGs $\mathcal{U}_1, \ldots, \mathcal{U}_M$ of $M$ entities
2: **Output:** Partition of $[M]$ into clusters
3: Let $S$ denote a uniform sample of $\frac{4\log(M/\delta)}{(\alpha-\beta)^2}$ nodes from $V$ selected with replacement.
4: **for** every entity $i \in [M]$ and $u \in S$ **do**
5:     $\text{ch}_i(u) \leftarrow$ IDENTIFY-OUTNBR$(\mathcal{U}_i, u)$
6:     $\text{sp}_i(u) \leftarrow$ IDENTIFY-BIDIRECTED$(\mathcal{U}_i, u)$
7:     $\text{pa}_i(u) \leftarrow \Gamma_i(u) \setminus (\text{ch}_i(u) \cup \text{sp}_i(u))$
8:     Construct $N_i(u)$ (defined in (2))
9: **end for**
10: Let $\mathcal{P}$ denote an empty graph on set of entities $[M]$
11: **for** every pair of entities $i, j$ **do**
12:     Let $\text{COUNT}(i,j) = \sum_{u \in S} \mathbf{1}\{N_i(u) = N_j(u)\}$
13:     **if** $\text{COUNT}(i,j) \geq \left(1 - \frac{\alpha+\beta}{2}\right)|S|$ **then**
14:         Include an edge between $i$ and $j$ in $\mathcal{P}$
15:     **end if**
16: **end for**
17: Return connected components in $\mathcal{P}$

---

used by an entity for this step is at most $\Delta \cdot |S|$. Combining all the above, using the next lemma, we show that with high probability Algorithm $(\alpha, \beta)$-BOUNDEDDEGREE recovers all the true clusters.

**Lemma 3.3.** *If the underlying MAGs $\mathcal{M}_1, \ldots, \mathcal{M}_M$ satisfy $(\alpha, \beta)$-clustering property with true clusters $C_1^\star, \ldots, C_k^\star$ and have maximum undirected degree $\Delta$. Then, the Algorithm $(\alpha, \beta)$-BOUNDEDDEGREE recovers the clusters $C_1^\star, \ldots, C_k^\star$ with probability at least $1 - \delta$. Every entity $i \in [M]$ uses at most $4(\Delta + 1)\log(M/\delta)/(\alpha - \beta)^2$ many atomic interventions.*

### 3.2 Learning Causal Graphs from $(\alpha, \beta)$-Clustering

In this section, we outline an approach to recover a close approximation of the causal MAGs of all the entities, after correctly recovering the clusters using Algorithm $(\alpha, \beta)$-BOUNDEDDEGREE. First, we note that since the $(\alpha, \beta)$-clustering allows the MAGs even in the same cluster to be different, the problem of exactly learning all the MAGs is challenging (with a small set of interventions) as causal edges learnt for an entity may not be relevant for another entity in the same cluster.

In the scenarios mentioned in the introduction, we expect the clusters to be more homogeneous, with many entities in the same cluster sharing the same MAG. We provide an overview of Algorithm $(\alpha, \beta)$-RECOVERY that recovers one such MAG called *dominant MAG* for every cluster. Consider a recovered cluster $C_a^\star$, and a partitioning $S_a^1, S_a^2, \cdots$ of MAGs such that all MAGs in a partition $S_a^i$ are equal for all $i$. We call the MAG $\mathcal{M}_a^{\text{dom}}$ corresponding to the largest partition $S_a^{\text{dom}}$ as the *dominant MAG* of $C_a^\star$. The dominant MAG of a cluster is parameterized by $\gamma_a = |S_a^{\text{dom}}|/|C_a^\star|$ (fraction of the MAGs in the cluster that belong to the largest partition). We defer additional details of Algorithm $(\alpha, \beta)$-RECOVERY to Appendix C.1.

**Overview of Algorithm $(\alpha, \beta)$-RECOVERY.** After recovering the clustering using Algorithm $(\alpha, \beta)$-BOUNDEDDEGREE, our goal is to learn the causal graphs. Using Algorithm $(\alpha, \beta)$-RECOVERY, we show that we can learn these graphs approximately up to a distance approximation of $\beta n$.

In a cluster $C_a^\star$, we construct a partitioning of MAGs such that two MAGs belong to a partition if they are equal. The MAG corresponding to the largest partition is called the *dominant MAG*. Using our algorithm, we learn the dominant MAG correctly and return it as an output. As all the MAGs in the cluster satisfy $(\alpha, \beta)$-clustering property, the dominant MAG is within a distance of $\beta n$ from the true MAG and therefore is a good approximation of the true MAG.

For learning the dominant MAG, there are two steps. First, we select a node uniformly at random for every entity and intervene on the node and its neighbors to learn all the edges incident on the node. Next, we construct the dominant MAG by combining the neighborhoods of each individual node. Let $u$ be any node and $T_u$ denote the set of all entities which intervened on $u$ in the first step. Now, among all the neighborhoods identified by the entities in $T_u$, we do not know which of them correspond to

that of the dominant MAG. In order to identify this, we use a threshold-based approach and assign a score to every entity in $T_u$. The score of an entity $i$ is the number of entities in $T_u$ that has the same neighborhood of $u$ as that of entity $i$. Finally, we select the entity with the maximum score and assign the neighborhood of the entity as the neighborhood of $u$ for the dominant MAG (Lines 12-15 in Algorithm $(\alpha, \beta)$-RECOVERY). We argue that if the cluster size is large (see Theorem 3.4), the neighborhoods of nodes using entities with maximum scores are equal to that of the dominant MAG. This is because the dominant MAG has the largest partition size, and if a sufficiently large number of entities (across all partitions) are assigned node $u$, then, many of them will be entities from the dominant MAG partition.

As the entities satisfy $(\alpha, \beta)$-clustering property, for all entities the recovered MAGs (dominant MAGs) are close to the true MAGs, and within a distance of at most $\beta n$. Note that any MAG from the cluster is within a distance of at most $\beta n$ due to $(\alpha, \beta)$-clustering property, but naively generating a valid MAG from a cluster will require $n$ interventions on one entity from Proposition 3.2. Our actual guarantee is somewhat stronger, as in fact, for the entities whose MAGs are dominant in their cluster, we do recover the exact MAGs. We have the result:

**Theorem 3.4.** *Suppose $\mathcal{M}_1, \mathcal{M}_2, \cdots \mathcal{M}_M$ satisfy $(\alpha, \beta)$ clustering property. If $\gamma_a > 1/2$ and $C_a^\star = \Omega(n \log(n/M\delta)(2\gamma_a - 1)^2)$ for all $a \in [k]$, then, Algorithm $(\alpha, \beta)$-RECOVERY recovers graphs $\widehat{\mathcal{M}}_1, \cdots \widehat{\mathcal{M}}_M$ such that for every entity $i \in [M]$, we have $d(\mathcal{M}_i, \widehat{\mathcal{M}}_i) \leq \beta n$ with probability $1 - \delta$. Every entity uses at most $(\Delta + 1) + 4(\Delta + 1) \log(M/\delta)/(\alpha - \beta)^2$ many atomic interventions.*

# 4 Causal Discovery under $\alpha$-Clustering Property

In the previous section, we discussed the more general $(\alpha, \beta)$-clustering scenario where we manage to construct a good approximation to all the MAGs. Now, we show that we can in fact recover all the MAGs exactly, if we make a stronger assumption. Suppose the MAGs $\mathcal{M}_1, \ldots, \mathcal{M}_M$ of the $M$ entities satisfy the $\alpha$-clustering property (Defn. 2.3). Firstly, we can design an algorithm similar to Algorithm $(\alpha, \beta)$-BOUNDEDDEGREE (see Algorithm $\alpha$-BOUNDEDDEGREE, Appendix D.3) that recovers the causal MAGs exactly with $O(\Delta \log(M/\delta)/\alpha)$ many interventions per entity, succeeding with probability $1 - \delta$. Note that this has a better $1/\alpha$ term in the intervention bound, instead of $1/\alpha^2$ (when $\beta = 0$) term arising in Theorem 3.4. In absence of latents, we can further improve it to $O(\log(M/\delta)/\alpha)$ many interventions per entity (see Algorithm NOLATENTS, Appendix D.2).

In this section, we present another approach (Algorithm $\alpha$-GENERAL) with an improved result that requires fewer number of interventions, even when $\Delta$ is big, provided that each cluster has at least $\Omega(n \log(M/\delta))$ entities. Missing details of Algorithm $\alpha$-GENERAL are in Appendix D.4.

**Overview of Algorithm $\alpha$-GENERAL.** First, using a similar approach as Algorithm $(\alpha, \beta)$-BOUNDEDDEGREE, we construct a uniform sample $S \subseteq V$, and find all the outgoing neighbors of nodes in $S$, for every entity $i \in [M]$. Then, we construct a graph on entities denoted by $\mathcal{P}$, where we include an edge between a pair of entities if the outgoing neighbors of the set of sampled nodes $S$, and the set of neighbors in PAGs associated with the entities (obtained from FCI) are the same. However, due to the presence of bidirected edges, it is possible that the connected components of $\mathcal{P}$ may not represent the true clusters $C_1^\star, \ldots, C_k^\star$.

We make the observation that a pair of entities $i, j$ that have an edge in this $\mathcal{P}$ and from different true clusters, can differ only if there is a node $u$ such that $u$ has a bidirected edge $u \leftrightarrow v$ in $\mathcal{M}_i$, and a directed edge $u \leftarrow v$ in $\mathcal{M}_j$ (or vice-versa). Intervening on both $u$ and $v$ will separate these entities, our main idea is to ensure that this happens. First, we show how to *detect* if there are at least two true clusters in any connected component of $\mathcal{P}$. Then, we identify all the entities belonging to these two clusters and remove the edges between these entities in $\mathcal{P}$ and continue.

More formally, let $T_1, \ldots, T_{k'}$ be the partition of $[M]$ provided by the $k'$ connected components of $\mathcal{P}$ and some of these can contain more than one true cluster, hence $k' \leq k$ and we focus on detecting such events. Let $\pi : [M] \to V$ denote a mapping from the set of entities to the nodes in $V$ such that $\pi(i)$ is chosen uniformly at random from $V$ for every entity $i$. For every entity $i$, we intervene on the node $\pi(i)$. To detect that there are at least two clusters in a given subset $T_a$ of entities, we show that there are two entities $i, j$ with an edge in $\mathcal{P}$ and for some node $u \in S$, we can identify the neighbor $v \in \Gamma_i(u) \cap \Gamma_j(u)$ such that $u \leftrightarrow v$ is an edge in $\mathcal{M}_i$ and $u \leftarrow v$ is an edge in $\mathcal{M}_j$ (or vice-versa). As there are at least $\Omega(n \log(M/\delta))$ entities in each of these two true clusters in $T_a$, for some $i, j \in T_a$, we can show that $\pi(i) = \pi(j) = v$ with probability at least $1 - \delta$.

After detecting the event that a component $T_a$ of $\mathcal{P}$ contains entities from at least two different true clusters (say, $C_b^\star$ and $C_c^\star$) due to an edge $(u, v)$ as above, we intervene on $v$ for every entity in $T_a$. By intervening on $v$ (and $u \in S$), we can separate all entities in $T_a$ that belong to true clusters $C_b^\star$ and $C_c^\star$, and remove edges between such entity pairs from $\mathcal{P}$. We repeat this above procedure of refining $\mathcal{P}$. In each iteration, we will have removed all edges between every pair of entities belonging to at least two different true clusters. Since there are at most $k^2$ different true cluster pairs, after $k^2$ iterations the connected components remaining correspond to the true clusters (with high probability). This can be done without knowing the value of $k$, by checking whether the connected components in $\mathcal{P}$ change or not after each iteration of the above idea.

**Going from Clustering to MAGs.** The idea of going from clusters to MAGs is simple and based on distributing the interventions across the entities in the cluster. Since under $\alpha$-clustering, entities belonging to a cluster share the same MAG, combining the results is relatively simpler (see Appendix D.1). Combining the guarantees of $\alpha$-GENERAL and $\alpha$-BOUNDEDDEGREE, we have:

**Theorem 4.1.** *If MAGs $\mathcal{M}_1, \ldots, \mathcal{M}_M$ satisfy $\alpha$-clustering property with true clusters $C_1^\star, \ldots, C_k^\star$ such that $\min_{b \in [k]} |C_b^\star| = \Omega(n \log(M/\delta))$. Then, there is an algorithm that exactly learns all these MAGs with probability at least $1 - \delta$. Every entity $i \in [M]$ uses $\min \left\{ O(\Delta \log(M/\delta)/\alpha), O(\log(M/\delta)/\alpha + k^2) \right\}$ many atomic interventions.*

**Lower Bound on the Number of Interventions**. We now give a lower bound on the number of atomic interventions needed for every algorithm that recovers the true clusters on the MAGs $\mathcal{M}_1, \mathcal{M}_2, \cdots \mathcal{M}_M$. Since a lower bound under $\alpha$-clustering is also a lower bound under $(\alpha, \beta)$-clustering, we work with the $\alpha$-clustering property here. First, we show that to identify whether a given pair of entities $i, j$ belong to the same true cluster or not, every (randomized or deterministic) algorithm must make $\Omega(1/\alpha)$ interventions for both $i$ and $j$.

Our main idea here is to use the famous Yao's minimax theorem [Yao, 1977] to get lower bounds on randomized algorithms. Yao's theorem states that an *average case* lower bound on a deterministic algorithm implies a *worst case* lower bound on randomized algorithms. To show a lower bound using Yao's minimax theorem, we construct a distribution $\mu$ on MAG pairs and show that every deterministic algorithm requires $\Omega(1/\alpha)$ interventions for distinguishing a pair of MAGs drawn from $\mu$. The construction of this distribution is presented in Appendix D.5. We summarize the result:

**Theorem 4.2.** *Suppose the underlying MAGs $\mathcal{M}_1, \ldots, \mathcal{M}_M$ satisfy $\alpha$-clustering property. In order to recover the clusters with probability $2/3$, every (randomized or deterministic) algorithm requires $\Omega(1/\alpha)$ interventions for every entity in $[M]$.*

## 5 Experimental Evaluation

In this section, we provide an evaluation of our approaches on data generated from real and synthetic causal networks for learning MAGs satisfying $(\alpha, \beta)$-clustering property. We defer additional details, results, and evaluation for $\alpha$-clustering to Appendix E.

**Causal Networks.** We consider the following real-world Bayesian networks from the *Bayesian Network Repository* which cover a wide variety of domains: *Asia* (Lung cancer) (8 nodes, 8 edges), *Earthquake* (5 nodes, 4 edges), *Sachs* (Protein networks) (11 nodes, 17 edges), and *Survey* (6 nodes, 6 edges). For the synthetic data, we use Erdös-Rényi random graphs (10 nodes). We use the term "causal network" to refer to these ground-truth Bayesian networks.

**Parameters**. For each causal network, we start from the corresponding DAG, and generate $M$ MAGs (one for each entity) split into $k$ clusters that satisfy the $(\alpha, \beta)$-clustering property through random changes to the graph. We also randomly introduce two latents in each graph, and account for them in MAG constructions. For more details, refer Appendix E. We set number of entities $M = 40$, number of clusters $k = 2$, $\alpha = 0.60, \beta = 0.20$, and dominant MAG parameter $\gamma = 0.90$ for both the clusters. For the synthetic data generated using Erdös-Rényi model, we use $n = 10$, probability of edge $0.3$.

**Evaluation of Clustering.** First, we focus on recovering the clustering using Algorithm $(\alpha, \beta)$-BOUNDEDDEGREE. As a baseline, we employ the well-studied FCI algorithm based on purely observational data [Spirtes et al., 2000]. After recovering the PAGs corresponding to the MAGs using FCI, we cluster them by constructing a similarity graph (similar to $(\alpha, \beta)$-BOUNDEDDEGREE) defined on the set of entities (refer Appendix E for more details). For Algorithm $(\alpha, \beta)$-BOUNDEDDEGREE, we first construct a sample $S$, and perform various interventions based on the set $S$ for every entity

| Causal Network | FCI | | | $(\alpha, \beta)$-BOUNDEDDEGREE (Alg. 3) | | | Maximum # Interventions |
| | Precision | Recall | Accuracy | Precision | Recall | Accuracy | |
|---|---|---|---|---|---|---|---|
| *Earthquake* | $0.57 \pm 0.18$ | $0.94 \pm 0.013$ | $0.58 \pm 0.18$ | $0.78 \pm 0.24$ | $0.92 \pm 0.03$ | $0.77 \pm 0.23$ | 4 |
| *Survey* | $0.62 \pm 0.21$ | $0.94 \pm 0.013$ | $0.62 \pm 0.2$ | $0.64 \pm 0.23$ | $0.97 \pm 0.02$ | $0.63 \pm 0.23$ | 5 |
| *Asia* | $0.57 \pm 0.18$ | $0.94 \pm 0.013$ | $0.58 \pm 0.18$ | $0.92 \pm 0.14$ | $0.95 \pm 0.03$ | $0.91 \pm 0.14$ | 5 |
| *Sachs* | $0.52 \pm 0.12$ | $0.94 \pm 0.01$ | $0.52 \pm 0.12$ | $0.89 \pm 0.20$ | $0.96 \pm 0.02$ | $0.88 \pm 0.19$ | 6 |
| *Erdös-Rényi* | $0.62 \pm 0.21$ | $0.94 \pm 0.02$ | $0.62 \pm 0.21$ | $1.0 \pm 0.00$ | $0.95 \pm 0.02$ | $0.97 \pm 0.013$ | 6 |

Table 1: In this table, we present the precision, recall and accuracy values obtained by our Algorithm $(\alpha, \beta)$-BOUNDEDDEGREE and using FCI. Each cell includes the mean value along with the standard deviation computed over 10 runs. The last column represents the maximum number of interventions per entity including both Algorithms $(\alpha, \beta)$-BOUNDEDDEGREE and $(\alpha, \beta)$-RECOVERY.

to obtain the clusters. We also implemented another baseline algorithm (GREEDY) that uses interventions, based on a greedy idea that selects nodes to set $S$ in Algorithm $(\alpha, \beta)$-BOUNDEDDEGREE by considering nodes in increasing order of their degree in the PAGs returned by FCI. We use this ordering to minimize the no. of interventions as we intervene on every node in $S$ and their neighbors.

**Metrics**. We use the following standard metrics for comparing the clustering performance: *precision* (fraction of pairs of entities correctly placed in a cluster together to the total number of pairs placed in a cluster together), *recall* (fraction of pairs of entities correctly placed in a cluster together to the total number of pairs in the same ground truth clusters), and *accuracy* (fraction of pairs of entities correctly placed or not placed in a cluster to the total number of pairs of entities).

**Results.** In Table 1, we compare Algorithm $(\alpha, \beta)$-BOUNDEDDEGREE to FCI on the clustering results. For Algorithm $(\alpha, \beta)$-BOUNDEDDEGREE, we use a sample $S$ of size 1, and observe in Figure 8 (Appendix E), that this corresponds to about 3 interventions per entity. With increase in sample size, we observed that the results were either comparable or better. We observe that our approach leads to considerably better performance in terms of the accuracy metric with an average difference in mean accuracy of about $0.25$. This is because FCI recovers partial graphs, and clustering based on the partial information results in poor accuracy. Because of the presence of a dominant MAG with in each cluster, we observe that the corresponding entities are always assigned to the same cluster, resulting in high recall for both $(\alpha, \beta)$-BOUNDEDDEGREE and FCI. We observe a higher value of precision for our algorithms, because FCI is unable to correctly classify the MAGs that are different from the dominating MAG.

Algorithm $(\alpha, \beta)$-BOUNDEDDEGREE outperforms the GREEDY baseline for the same sample(S) size. For example, on the *Earthquake* and *Survey* causal networks, Algorithm $(\alpha, \beta)$-BOUNDEDDEGREE obtains the mean accuracy values of $0.77$ and $0.63$ respectively, while GREEDY for the same number of interventions obtained an accuracy of only $0.487$ and $0.486$ respectively. For the remaining networks, the accuracy values of GREEDY are almost comparable to our Algorithm $(\alpha, \beta)$-BOUNDEDDEGREE.

After clustering, we recover the dominant MAGs using Algorithm $(\alpha, \beta)$-RECOVERY, and observe that the additional interventions needed are bounded by the maximum degree of the graphs (see Theorem 3.4). This is represented in the last column in Table 1. We observe that our *collaborative* algorithms use fewer interventions for dominant MAG recovery compared to the number of nodes in each graph. E.g., in the Erdös-Rényi setup, the number of nodes $n = 10$, whereas we use at most 6 interventions per entity. Thus, compared to the worst-case, cutting the number of interventions for each entity by $40\%$.

# 6 Conclusion

We introduce a new model for causal discovery to capture practical scenarios where are multiple entities with different causal structures. Under natural clustering assumption(s), we give efficient provable algorithms for causal learning with atomic interventions and demonstrate its empirical performance. Our model can be extended to the setting where all interventions are non-adaptive, and we plan to study it as part of future work. An interesting future direction would be to use interventional equivalence classes of DAGs as part of the model, instead of the clustering assumption. This might require extending the interventional equivalence between DAGs studied in [Hauser and Bühlmann, 2012, Katz et al., 2019] to the setting without the causal sufficiency assumption and exploit that for learning.

## Acknowledgements

The work was partially supported by NSF grants 1934846, 1908849, 1637536 and an Adobe Faculty Research grant.

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
