# Appendix for "Collaborative Causal Discovery with Atomic Interventions"

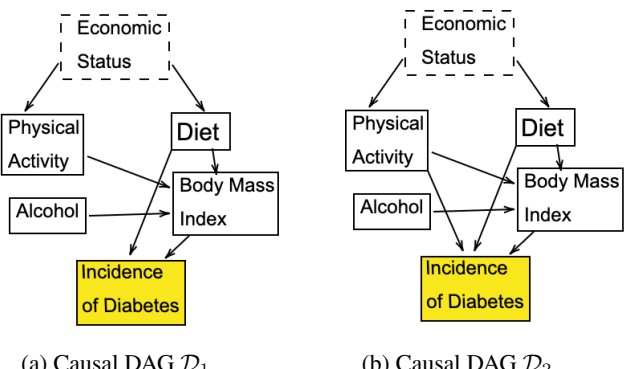

(a) Causal DAG $\mathcal{D}_1$      (b) Causal DAG $\mathcal{D}_2$

Figure 3: Two possible diabetes incidence graphs for an individual from [Joffe et al., 2012] differing in the causal edge between *Physical Activity* and *Incidence of Diabetes*. The observed variables include: *Diet, Body Mass Index (BMI), Physical Activity, Alcohol (consumption), Incidence of Diabetes*, and the unobserved variable (latent) is *Economic Status*. The variable *Incidence of Diabetes* is observable but can't be intervened on, this is not an issue as it has no outgoing edges in the graphs. In this paper, we do not know the underlying causal graphs or which individuals share the same graph. As intervening on variables such as *Diet, BMI* might need expensive and careful experimental organization, we ask the following question – given a collection of independent entities (in this diabetes example, they can refer to a collection of people), can we collaboratively learn each entity's causal graphs while minimizing the number of interventions per entity?

## A  Missing Details from Section 2

### A.1  Maximal Ancestral Graphs

Ancestral graphical models were introduced motivated by the need to represent data generating processes that involve latent variables. In this paper, we work with a class of graphical models, the maximal ancestral graph (MAG), which are a generalization of DAGs and are closed under marginalization and conditioning [Richardson and Spirtes, 2002]. A maximal ancestral graph (MAG) is a (directed) mixed graph that may contain two kinds of edges: directed edges ($\rightarrow$) and bi-directed edges ($\leftrightarrow$). Before defining a MAG, we need some preliminaries.

Consider a mixed graph $\mathcal{G}$. Given an path $\pi = \langle u, \ldots, w, \ldots, v \rangle$, $w$ is a collider on $\pi$ if the two edges incident to $w$ in $\pi$ are both into $w$, that is, have an arrowhead into $w$; otherwise it is called a non-collider on $\pi$. Let $S$ be any subset of nodes in the graph $\mathcal{G}$. An *inducing path* relative to $S$ is a path on which every node not in $S$ (except for the endpoints) is a collider on the path and every collider is an ancestor of an endpoint of the path.

**Definition A.1.** *A mixed graph is called a maximal ancestral graph (MAG) if*

1. *The mixed graph is* ancestral, *i.e., it has no directed cycles, and whenever there is a bidirected edge $u \leftrightarrow v$, then there is no directed path from $u$ to $v$ or $v$ to $u$.*

2. *There is no inducing path between any two non-adjacent nodes.*

It is straightforward to extend the notion of d-separation in DAGs to mixed graphs using the notion of m-separation  [Richardson and Spirtes, 2002].

**Definition A.2.** *In a mixed graph, a path $\pi$ between nodes $u$ and $v$ is m-connecting relative to a (possibly empty) set of nodes $Z$ with $u, v \notin Z$ if*

1. *every non-collider on $\pi$ is not a member of $Z$;*

2. *every collider on $\pi$ is an ancestor of some member of $Z$.*

*$u$ and $v$ are said to be m-separated by $Z$ if there is no m-connected path between $u$ and $v$ relative to $Z$.*

**Conversion of a DAG to a MAG.** The following construction gives us a MAG $\mathcal{M}$ from a DAG $\mathcal{D}$:

1. for each pair of variables $u, v \in V$, $u$ and $v$ are adjacent in $\mathcal{M}$ if and only if there is an inducing path between them relative to $L$ in $\mathcal{D}$. The skeleton or the undirected graph constructed from PAG $\mathcal{U}$ (obtained using FCI [Spirtes et al., 2000]) by ignoring the directions of edges captures all the edges in $\mathcal{M}$.

2. for each pair of adjacent variables $u, v$ in $\mathcal{M}$, orient the edge as $u \rightarrow v$ in $\mathcal{M}$ if $u$ is an ancestor of $v$ in $\mathcal{D}$; orient it as $u \leftarrow v$ in $\mathcal{M}$ if $v$ is an ancestor of $u$ in $\mathcal{D}$; orient it as $u \leftrightarrow v$ in $\mathcal{M}$ otherwise.

Several DAGs can lead to the same MAG (See Figure 4c). Essentially a MAG represents a set of DAGs that have the exact same d-separation structures and ancestral relationships among the observed variables. By construction, the MAG is unique for a given DAG.

As a further evidence to the claim that interventions are required, see Figure 5, that gives an example of two MAGs separated by a distance of $\frac{n}{2}$ and have the same Partial Ancestral Graph identified by FCI [Zhang, 2008b].

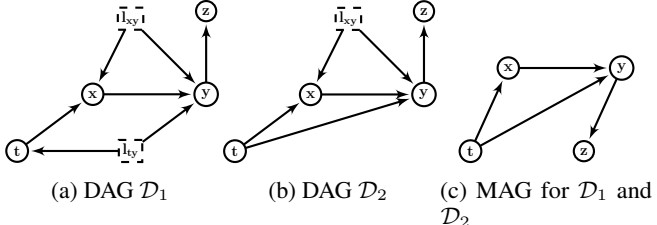

(a) DAG $\mathcal{D}_1$       (b) DAG $\mathcal{D}_2$       (c) MAG for $\mathcal{D}_1$ and $\mathcal{D}_2$

Figure 4: Different DAGs with same MAG. It is easy to observe that, no single vertex interventions can differentiate $\mathcal{D}_1$ from $\mathcal{D}_2$.

**Conditional Independence (CI) Tests**. Conditional independence tests are an important building block in causal discovery.

(i) CI-test in observational distribution: Given $u, v \in V$, $Z \subset V$ check whether $u$ is independent of $v$ given $Z$, denoted by $u \perp\!\!\!\perp v \mid Z$.

(ii) CI-test in interventional distribution: Given $u, v \in V$, $Z \subset V$, and $w \in V$, check whether $u$ is independent of $v$ given $Z$ in the interventional distribution of $w$, denoted by $u \perp\!\!\!\perp v \mid Z, \mathrm{do}(w)$ where $\mathrm{do}(w)$ is the intervention on the variable $w$.

The convergence rates of CI tests are well-known [Neykov et al., 2021] which can be used to obtain the required sample size bounds for any of the PAG estimation procedures for the desired Type 1 error bound (omitted here). Note that in our experiments (Section 5), we do run CI tests on actual data samples generated by our model.

# B    Helper Routines

**Claim B.1.** *Suppose $\mathcal{D}_i$ is the DAG and $\mathcal{M}_i$ is the corresponding MAG for some entity $i \in [M]$. Then, $u \not\perp\!\!\!\perp v \mid \mathrm{do}(u)$ iff $u$ is an ancestor of $v$ in the graph $\mathcal{D}_i$.*

*Proof.* We follow a proof similar to Lemma 1 in [Kocaoglu et al., 2017]. If $u$ is an ancestor of $v$ in the graph $\mathcal{D}_i$ using the path $\pi_{uv}$, then, in the mutilated graph corresponding to $\mathrm{do}(u)$, the path $\pi_{uv}$ remains intact. From d-separation [Pearl, 2009], $\pi_{uv}$ can only be blocked by conditioning on one of the nodes that are not end points. As we do not condition on any variables in the CI-test $u \perp\!\!\!\perp v \mid \mathrm{do}(u)$ and therefore do not block the path $\pi_{uv}$, we have $u \not\perp\!\!\!\perp v \mid \mathrm{do}(u)$.

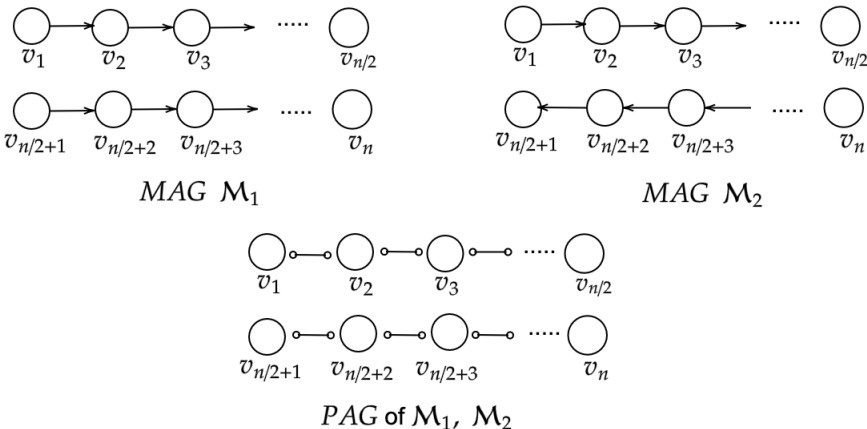

Figure 5: An example of MAGs $\mathcal{M}_1$ and $\mathcal{M}_2$ with large distance $d(\mathcal{M}_1, \mathcal{M}_2)$ but generating the same PAG.

Now, we consider the other direction. If $u \not\perp\!\!\!\perp v \mid \mathrm{do}(u)$, then, there is at least a path $\pi_{uv}$ between $u$ and $v$ that is not blocked. In the mutilated graph corresponding to the interventional distribution $\mathrm{do}(u)$, the incoming edges into the node $u$ are removed. In the path $\pi_{uv}$, the edge incident on $u$ is an outgoing edge. If there is a collider on $\pi_{uv}$, we have blocked the path by not conditioning on it (from d-separation). As the path is not blocked, it implies that there is no collider on the path. Therefore, the path $\pi_{uv}$ is a directed path from $u$ to $v$. Hence, the claim. $\qquad\square$

**Claim B.2.** *Given an entity $i \in [M]$, and a node $u \in V$, Algorithm* IDENTIFY-OUTNBR *identifies all outgoing edges of $u$ in $\mathcal{M}_i$ ($ch_i(u)$) correctly using an intervention on $u$.*

*Proof.* We know that $\mathcal{U}_i = (V, \widehat{E}_i)$ represents the partial ancestral graph of $\mathcal{M}_i$. We observe that any outgoing edge $(u, v)$ incident on a node $u$ in the PAG $\mathcal{U}_i$ can be of the form $u\circ\!\!-\!\!\circ v$ or $u\circ\!\!\rightarrow v$. Otherwise, we already know that the edge is not an outgoing edge from $u$. We claim that we can identify an outgoing edge $(u, v)$ from a node $u$ correctly, if CI-test returns $u \not\perp\!\!\!\perp v \mid \mathrm{do}(u)$ for every $v \in \Gamma_i(u)$ satisfying the condition mentioned above. From Claim B.1, we have that $u \not\perp\!\!\!\perp v \mid \mathrm{do}(u)$ iff $u$ is an ancestor of $v$ in $\mathcal{D}_i$, which implies $u \rightarrow v$ is present in $\mathcal{M}_i$ and $v \in \mathrm{ch}_i(u)$. $\qquad\square$

**Claim B.3.** *Given an entity $i \in [M]$, and a node $u \in V$, Algorithm* IDENTIFY-BIDIRECTED *identifies all bidirected edges incident on $u$ in $\mathcal{M}_i$ ($sp_i(u)$) correctly using atomic interventions on all nodes in $\Gamma_i(u)$.*

*Proof.* We observe that any bi-directed edge $(u, v)$ incident on a node $u \in V$ in the PAG $\mathcal{U}_i$ can be of the form $u\circ\!\!-\!\!\circ v$ or $u\leftarrow\!\!\circ v$ or $u\circ\!\!\rightarrow v$. Otherwise, we already know that the edge is not a bi-directed edge incident at $u$. In Algorithm IDENTIFY-BIDIRECTED, for every neighbor $v$ of $u$ in the PAG $\mathcal{U}_i$ satisfying the above condition, we check if $u \perp\!\!\!\perp v \mid \mathrm{do}(u)$ and $v \perp\!\!\!\perp u \mid \mathrm{do}(v)$ is satisfied. From Claim B.1, we know that if $u \not\perp\!\!\!\perp v \mid \mathrm{do}(u)$, then $u$ is an ancestor of $v$ in $\mathcal{D}_i$ (similarly, $v$ is an ancestor of $u$ in $\mathcal{D}_i$ if $u \not\perp\!\!\!\perp v \mid \mathrm{do}(v)$). So, if $u \perp\!\!\!\perp v \mid \mathrm{do}(u)$ and $u \perp\!\!\!\perp v \mid \mathrm{do}(v)$, then, $u$ is not an ancestor of $v$ or vice-versa, which implies $u \leftrightarrow v$ is present in $\mathcal{M}_i$, i.e., $v \in \mathrm{sp}_i(u)$. As we perform an intervention for every neighbor of $u$ in $\mathcal{U}_i$, we have the claim.

$\qquad\square$

**Algorithm RECOVERG.** For every $u \in V$, first identify outgoing neighbors using Algorithm IDENTIFY-OUTNBR and then identify all the bidirected edges incident on $u$ using Algorithm IDENTIFY-BIDIRECTED.

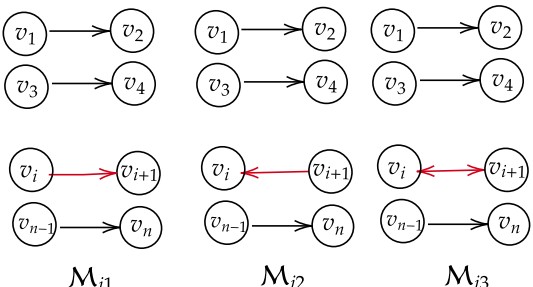

Figure 6: The MAGs used in the proof of Proposition B.5.

**Lemma B.4.** *Algorithm* RECOVERG *recovers all edges of* $\mathcal{M}_i$, *for an entity* $i \in [M]$ *using* $n$ *atomic interventions.*

*Proof.* Given an entity $i \in [M]$, we obtain the partial ancestral graph $\mathcal{U}_i$ from observational data. Using Algorithm RECOVERG, we create interventions for every node $u \in V$. For every node $u$, we correctly identify all the outgoing neighbors of $u$ using Algorithm IDENTIFY-OUTNBR (Claim B.2) and all the bidirected edges using Algorithm IDENTIFY-BIDIRECTED (Claim B.3). Therefore, we have recovered all edges of $\mathcal{M}_i$ using $n$ atomic interventions. □

**Proposition B.5.** *[Proposition 3.2 restated] There exists a causal MAG* $\mathcal{M}$ *such that every adaptive or non-adaptive algorithm requires* $n$ *many atomic interventions to recover* $\mathcal{M}$.

*Proof.* Suppose the set of nodes of an unknown MAG $\mathcal{M}$ is given by $V = \{v_1, v_2, \cdots v_n\}$. We denote ALG by any adaptive or non-adaptive *deterministic* algorithm that recovers $\mathcal{M}$ using the set of interventions $\mathcal{S} \subseteq V$. For the sake of contradiction, let ALG recover $\mathcal{M}$ correctly and $v_i$ be the vertex that has not been intervened on, i.e., $v_i \notin \mathcal{S}$.

Construct the MAGs $\mathcal{M}_{i1}, \mathcal{M}_{i2}, \mathcal{M}_{i3}$ with edges $E_{i1} = \{v_1 \rightarrow v_2, v_3 \rightarrow v_4, \cdots, v_i \rightarrow v_{i+1}, \cdots, v_{n-1} \rightarrow v_n\}, E_{i2} = \{v_1 \rightarrow v_2, v_3 \rightarrow v_4, \cdots, v_i \leftarrow v_{i+1}, \cdots, v_{n-1} \rightarrow v_n\}, E_{i3} = \{v_1 \rightarrow v_2, v_3 \rightarrow v_4, \cdots, v_i \leftrightarrow v_{i+1}, \cdots, v_{n-1} \rightarrow v_n\}$ respectively (see Figure 6).

Upon termination, ALG will have recovered one of the MAGs $\mathcal{M}_{i1}$, $\mathcal{M}_{i2}$ or $\mathcal{M}_{i3}$. As $v_i \notin \mathcal{S}$, we will argue that the true MAG is different from the recovered MAG. We consider two cases:

1. If $v_{i+1} \in \mathcal{S}$. First, we observe that for all three MAGs $\mathcal{M}_{i1}, \mathcal{M}_{i2}$ and $\mathcal{M}_{i3}$, the CI-test $v_i \not\perp\!\!\!\perp v_{i+1}$. For MAGs $\mathcal{M}_{i1}$ and $\mathcal{M}_{i2}$, we have $v_i \perp\!\!\!\perp v_{i+1} \mid \mathrm{do}(v_{i+1})$ while $v_i \not\perp\!\!\!\perp v_{i+1} \mid \mathrm{do}(v_{i+1})$ for MAG $\mathcal{M}_{i2}$. As these are the only possible CI-tests for vertices $v_i$ and $v_{i+1}$, the algorithm ALG cannot differentiate between $\mathcal{M}_{i1}$ and $\mathcal{M}_{i3}$. If ALG recovers $\mathcal{M}_{i1}$, then, we can set $\mathcal{M}$ to be $\mathcal{M}_{i3}$. This is a contradiction.

2. If $v_{i+1} \notin \mathcal{S}$. We observe that for all three MAGs $\mathcal{M}_{i1}, \mathcal{M}_{i2}$ and $\mathcal{M}_{i3}$, the CI-test $v_i \not\perp\!\!\!\perp v_{i+1}$, and it is the only possible CI-test involving vertices $v_i$ and $v_{i+1}$. Therefore, the algorithm ALG cannot differentiate between $\mathcal{M}_{i1}, \mathcal{M}_{i2}$ and $\mathcal{M}_{i3}$. If ALG recovers $\mathcal{M}_{i1}$, then, we can set $\mathcal{M}$ to be $\mathcal{M}_{i2}, \mathcal{M}_{i3}$ and similarly for other cases. This is a contradiction.

Therefore, to recover $\mathcal{M} \in \{\mathcal{M}_{i1}, \mathcal{M}_{i2}, \mathcal{M}_{i3}\}$ correctly, we must have $v_i \in \mathcal{S}$. As $i$ is chosen arbitrarily, and for every $i$ we can construct the MAGs $\mathcal{M}_{i1}, \mathcal{M}_{i2}, \mathcal{M}_{i3}$, such that any adaptive or non-adaptive deterministic algorithm requires interventions on every node.

We can extend the proof to include *randomized* algorithms, with success probability strictly greater than $1/2$, by observing that when $v_i \notin \mathcal{S}$, ALG has at least two MAGs among $\mathcal{M}_{i1}, \mathcal{M}_{i2}, \mathcal{M}_{i3}$ that it cannot differentiate (as argued using two cases above).

□

### B.1 Handling Uncertainty in PAG Estimation

We assume throughout this paper that the initial PAGs (fed to our algorithms) are estimated correctly from observational data. We outline some reasons behind such an assumption.

(a) PAG estimation is a very well-studied problem in causal discovery from both a theoretical and practical perspective. Well known algorithms for recovering PAGs, such as FCI (Fast Causal Inference), are known to be sound and complete (see [Spirtes et al., 1999] and [Zhang, 2008b]). Also, recent variations of FCI such as Really Fast Causal Inference (RFCI) have sped up the FCI procedure [Colombo et al., 2012]. Today FCI/RFCI procedures are commonly used in practice, with various implementations available [Kalisch et al., 2012].

(b) Note, for all our algorithms and bounds, all that we require from the PAGs is that they have the correct (undirected) skeleton as their corresponding MAGs, i.e., we could just ignore all the directed edges in the initial PAGs and replace them with edges before using them in our algorithms, and this would not change our results.

(c) Finally, we could even relax our assumptions and tolerate error even in skeleton estimation. The idea is simple, and we sketch it here. Suppose the MAGs $\mathcal{M}_1, \mathcal{M}_2, \cdots \mathcal{M}_M$ satisfy the $\alpha$-clustering assumption with true clusters $C_1^\star, C_2^\star, \cdots C_k^\star$. Now consider the setting where we have errors in the PAG skeleton estimation. Let $\mathcal{U}_1, \mathcal{U}_2, \cdots \mathcal{U}_M$ be the true skeletons of the MAGs $\mathcal{M}_1, \mathcal{M}_2, \cdots \mathcal{M}_M$. Consider for each MAG $\mathcal{M}_i$, a corrupted counterpart $\mathcal{M}_i^{\text{corr}}$, with the guarantee that $d(\mathcal{M}_i, \mathcal{M}_i^{\text{corr}}) \leq \beta/2n$. These corrupted MAGs are only constructed for the sake of proof, and are not actually present. Assume that the skeleton estimation is not precise and instead of $\mathcal{U}_1, \mathcal{U}_2, \cdots \mathcal{U}_M$, it produces the skeletons $\mathcal{U}_1^{\text{corr}}, \mathcal{U}_2^{\text{corr}}, \cdots \mathcal{U}_M^{\text{corr}}$, associated with these corrupted MAGs $\mathcal{M}_1^{\text{corr}}, \mathcal{M}_2^{\text{corr}}, \cdots \mathcal{M}_M^{\text{corr}}$. By triangle inequality, it is easy to observe that the MAGs satisfy $(\alpha - \beta, \beta)$-clustering assumption. If $\beta < \alpha/2$, then, using Algorithm $(\alpha, \beta)$-BOUNDEDDEGREE on $\mathcal{U}_1^{\text{corr}}, \mathcal{U}_2^{\text{corr}}, \cdots \mathcal{U}_M^{\text{corr}}$ with parameter $\alpha$ replaced by $\alpha - \beta$ will guarantee that we recover the true clusters $C_1^\star, C_2^\star, \cdots C_k^\star$. This follows because any pair of entities $i, j$ that were originally in the same true cluster will still remain together in the same cluster, even under corruption, as their corrupted MAGs will be at most $\beta n < \alpha/2n$ distance apart. Similarly, if $i, j$ belonged to different true clusters then they will still remain in different clusters, even under corruption, as their corrupted MAGs will be $> \alpha/2n$ distance apart. Also, if the corrupted MAGs satisfy the conditions in Theorem 3.4, we can recover the dominant MAG. With the right set of parameters, this argument can also be extended starting from an $(\alpha, \beta)$-clustering.

## C  Discovery under $(\alpha, \beta)$-Clustering

In this section, we present an algorithm that recovers the underlying clusters $C_1^\star, C_2^\star, \cdots, C_k^\star$ provided they satisfy $(\alpha, \beta)$-clustering property. After recovering the clusters, in Section C.1, we give an algorithm that recovers an approximate MAG for every entity with only few additional interventions.

Firstly, using the next lemma, we show that the threshold used by Algorithm $(\alpha, \beta)$-BOUNDEDDEGREE correctly identifies whether two entities belong to the same true cluster or not. This implies that our algorithm $(\alpha, \beta)$-BOUNDEDDEGREE recovers the clusters with high probability.

**Lemma C.1** (Lemma 3.3 Restated). *If the underlying MAGs $\mathcal{M}_1, \ldots, \mathcal{M}_M$ satisfy $(\alpha, \beta)$-clustering property with true clusters $C_1^\star, \ldots, C_k^\star$ and have maximum undirected degree $\Delta$. Then, the Algorithm $(\alpha, \beta)$-BOUNDEDDEGREE recovers the clusters $C_1^\star, \ldots, C_k^\star$ with probability at least $1 - \delta$. Every entity $i \in [M]$ uses at most $4(\Delta + 1)\log(M/\delta)/(\alpha - \beta)^2$ many atomic interventions.*

*Proof.* Let $\text{COUNT}(i, j) = \sum_{u \in S} \mathbf{1}\{N_i(u) = N_j(u)\}$ for distinct entities $i, j$. If $i, j$ belong to the same true cluster $C_t^\star$ for some $t \in [k]$, we have :

$$\mathbf{E}[\text{COUNT}(i, j)] = \mathbf{E}\left[\sum_{u \in S} \mathbf{1}\{N_i(u) = N_j(u)\}\right] \geq (1 - \beta)|S|$$

Using Hoeffding's inequality, with probability at least $1 - \exp\left(-\frac{\Lambda^2}{2|S|}\right)$

$$\text{COUNT}(i, j) \geq \mathbf{E}[\text{COUNT}(i, j)] - \frac{\Lambda}{2}$$

If $i, j$ belong to different true clusters, then, we have :

$$\mathbf{E}[\text{COUNT}(i,j)] = \mathbf{E}\left[\sum_{u \in S} \mathbf{1}\{N_i(u) = N_j(u)\}\right] \leq (1-\alpha)|S|$$

Using Hoeffding's inequality, with probability at least $1 - \exp\left(-\frac{\Lambda^2}{2|S|}\right)$

$$\text{COUNT}(i,j) < \mathbf{E}[\text{COUNT}(i,j)] + \frac{\Lambda}{2}$$

Set $\Lambda = |S|(\alpha - \beta)$ and $|S| = \frac{4 \log M/\delta}{(\alpha-\beta)^2}$.

Using union bound for every pair of entities in $[M]$, we have with probability at least $1 - \delta$:

if entities $i, j \in C_t^\star$ (belong to the same true cluster) : $\text{COUNT}(i,j) \geq \left(1 - \frac{\alpha+\beta}{2}\right)|S|$ and

if entities $i, j \notin C_b^\star \ \forall b \in [k]$ (do not belong to the same true cluster) : $\text{COUNT}(i,j) < \left(1 - \frac{\alpha+\beta}{2}\right)|S|$

Therefore, every pair of entities from same true cluster satisfy the condition that COUNT value is larger than $(1 - \frac{\alpha+\beta}{2})|S|$ and will include an edge in $\mathcal{P}$, while we do not include an edge between pair of entities from different clusters. The resulting graph $\mathcal{P}$, will have $k$ connected components and Algorithm $(\alpha, \beta)$-BOUNDEDDEGREE will return the true clusters correctly.

As we intervene on all the neighbors of every node in $S$, it will increase the interventions for every entity by a multiplicative $\Delta + 1$ factor. For an entity $i$, the total number of interventional distributions constructed is

$$\sum_{u \in S}(1 + |\Gamma_i(u)|) \leq |S|(\Delta + 1) = 4(\Delta+1)\log(M/\delta)/(\alpha-\beta)^2 \ \text{as} \ \max_{i \in [M], w \in V}|\Gamma_i(w)| \leq \Delta.$$

This completes the proof. □

## C.1 Learning Causal Graphs from Clusters

In this section, we give the Algorithm $(\alpha, \beta)$-RECOVERY that returns an approximate causal graph for every entity $i \in [M]$. We also include the brief overview from Section 3.2 for clarity.

**Overview of Algorithm $(\alpha, \beta)$-RECOVERY.** Consider a cluster $C_a^\star$. We recover the dominant MAG of this cluster, $\mathcal{M}_a^{\text{dom}}$, by recovering all the neighbors of every node and carefully merging them. Our idea is to assign a node, selected uniformly at random, to every entity in $C_a^\star$, and recover the neighborhood of the node using Algorithms IDENTIFY-OUTNBR and IDENTIFY-BIDIRECTED. If the clusters are large such that $|C_a^\star| \gg n$ (see Theorem 3.4 for a precise bound), we can show a large number of entities $T_u$ are assigned node $u$, and many of them will share the dominant MAG. We maintain a count $\text{NCOUNT}(i, u)$ of the number of times the entity $i$ agrees with other entities in $T_u$ about neighbors of $u$, and guarantee (with high probability) that the entity with the highest count will be that of dominant MAG. After merging the neighbors recovered for every node, we assign the resulting graph to every entity in the cluster.

---

**Algorithm 4** $(\alpha, \beta)$-RECOVERY

---

1: **Input:** $\alpha > 0$, $\beta \geq 0 \, (< \alpha)$, confidence parameter $\delta > 0$, PAGs $\mathcal{U}_1, \ldots, \mathcal{U}_M$ of $M$ entities
2: **Output:** $\widehat{\mathcal{M}}_1, \widehat{\mathcal{M}}_2, \cdots \widehat{\mathcal{M}}_M$ representing set of $M$ MAGs.
3: Obtain clusters $C_1^\star, C_2^\star, \cdots, C_k^\star$ using Algorithm 3.
4: **for** every cluster $C_a^\star$ where $a \in [k]$ **do**
5:     Let $\widehat{\mathcal{M}}_a^{\text{dom}}$ be an empty graph on the set of nodes $V$.
6:     For every entity $i \in C_a^\star$, select a node $u \in V$ uniformly at random and assign it to $u$ represented by the set $T_u$.
7:     **for** every node $u \in V$ **do**
8:         **for** every entity $i \in T_u$ **do**
9:             $\text{ch}_i(u) \leftarrow$ IDENTIFY-OUTNBR$(\mathcal{U}_i, u)$
10:             $\text{sp}_i(u) \leftarrow$ IDENTIFY-BIDIRECTED$(\mathcal{U}_i, u)$.
11:             $\text{pa}_i(u) \leftarrow \Gamma_i(u) \setminus (\text{ch}_i(u) \cup \text{sp}_i(u))$.
12:             Construct $N_i(u)$ (defined in (2)) and calculate $\text{NCOUNT}(i, u) = \sum_{j \in T_u : j \neq i} \mathbf{1}\{N_i(u) = N_j(u)\}$
13:         **end for**
14:         Let $u_{\max} \leftarrow \arg\max_{i \in T_u} \text{NCOUNT}(i, u)$.
15:         Set neighbors of $u$ in $\widehat{\mathcal{M}}_a^{\text{dom}}$ to the set $N_{u_{\max}}(u)$.
16:     **end for**
17:     For every entity $i \in C_a^\star$, set $\widehat{\mathcal{M}}_i = \widehat{\mathcal{M}}_a^{\text{dom}}$.
18: **end for**
19: Return $\widehat{\mathcal{M}}_1, \widehat{\mathcal{M}}_2, \cdots \widehat{\mathcal{M}}_M$

---

Consider the cluster $C_a^\star$ for some $a \in [k]$. In the next claim, we show that if the size of $C_a^\star$ is sufficiently large, then, each node $u \in V$ is assigned a large number of entities by $(\alpha, \beta)$-RECOVERY using the set $T_u$.

**Claim C.2.** *Consider a cluster $C_a^\star$ such that $\gamma_a > 1/2$ and $|C_a^\star| \geq \frac{8n \log(nM/\delta)}{(2\gamma_a - 1)^2}$. Let $T_u$ denote the set of entities assigned to node $u$ in Algorithm 4. Then, we have with probability $1 - \delta$, $|T_u| \geq \frac{4 \log(nM/\delta)}{(2\gamma_a - 1)^2}$ for every node $u \in V$.*

*Proof.* For a node $u \in V$, and cluster $C_a^\star$, we have:

$$\mathbf{E}[T_u] = \frac{|C_a^\star|}{n} \geq \frac{8 \log(nM/\delta)}{(2\gamma_a - 1)^2}.$$

Using Chernoff bound, with probability at least $1 - \exp\left(-\log(nM/\delta)/(2\gamma_a - 1)^2\right) \geq 1 - \delta/nM$, we have:

$$T_u \geq \frac{\mathbf{E}[T_u]}{2} \geq \frac{4 \log(nM/\delta)}{(2\gamma_a - 1)^2}.$$

Applying union bound for every node $u \in V$ and $a \in [k]$, gives us the claim. $\qquad\square$

Consider a partitioning of $C_a^\star$ given by $S_a^1, S_a^2, \cdots S_a^t$ where each $S_a^i$ for any $i \in [t]$ represents the maximal collection of MAGs that are equal. Formally, we have:

$$S_a^i = \{\mathcal{M}_p \mid \mathcal{M}_p \in C_a^\star \text{ and } \mathcal{M}_p = \mathcal{M}_q \quad \forall \mathcal{M}_q \in S_a^i\}.$$

Let $|S_a^{\text{dom}}| \geq |S_a^i|$ for every partition $i \in [t]$ and $\text{dom}_a$ denote an entity in $S_a^{\text{dom}}$. We define:

$$G_a(u) = \{j \mid j \in C_a^\star \text{ and } N_i(u) = N_{\text{dom}_a}(u)\} \text{ and } B_a(u) = C_a^\star \setminus G_a(u).$$

We can observe that:

$$|G_a(u)| \geq |S_a^{\text{dom}}| \text{ and } |B_a(u)| \leq |C_a^\star| - |S_a^{\text{dom}}|.$$

Conditioned on the previous claim that each set $T_u$ for all $u \in V$ is large, we argue that for any pair of entities $i, j \in C_a^\star$ where $\mathcal{M}_i = \mathcal{M}_a^{\text{dom}}$, and $\mathcal{M}_j \neq \mathcal{M}_a^{\text{dom}}$, the NCOUNT value calculated by $(\alpha, \beta)$-RECOVERY of entity $i$ for the node $u$ is always larger than that of entity $j$. Intuitively, after assigning the entities to nodes, we observe that for every node $u \in V$, the set $T_u$ contains a large number of entities with the dominant MAG, i.e., $|T_u \cap G_a(u)|$ is large. Because dominant MAGs share the same neighborhood (as they represent the same graph), we can show that the NCOUNT value of dominant MAG is larger than any other MAG in the cluster. We formalize this statement using the following lemma.

**Lemma C.3.** *For every $a \in [k]$, $u \in V$ and any pair of entities $i, j \in C_a^\star$ that satisfy $i \in G_a(u)$ and $j \in B_a(u)$, we have with probability $1 - \delta$,*

$$\text{NCOUNT}(i, u) > \text{NCOUNT}(j, u).$$

*Proof.* From Algorithm 4, we know that $\text{NCOUNT}(i, u) = \sum_{j \neq i, j \in T_u} \mathbf{1}\{N_i(u) = N_j(u)\}$ for an entity $i \in T_u$ and a node $u \in V$. Consider the case $i \in G_a(u)$. Then, we have:

$$\mathbf{E}[\text{NCOUNT}(i, u)] = \mathbf{E}\left[\sum_{j \neq i, j \in T_u} \mathbf{1}\{N_i(u) = N_j(u)\}\right]$$

$$= \mathbf{E}\left[\sum_{j \neq i, j \in T_u} \mathbf{1}\{N_{\text{dom}_a}(u) = N_j(u)\}\right]$$

$$= \mathbf{E}[|T_u \cap G_a(u)|]$$

$$\geq \frac{|S_a^{\text{dom}}|}{n} = \frac{|S_a^{\text{dom}}|}{|C_a^\star|} \cdot \frac{|C_a^\star|}{n} = \gamma_a \cdot \frac{|C_a^\star|}{n}$$

Using Hoeffding's inequality, with probability at least $1 - \exp\left(-\frac{\Lambda^2}{2|T_u|}\right)$

$$\text{NCOUNT}(i, u) \geq \mathbf{E}[\text{NCOUNT}(i, u)] - \frac{\Lambda}{2} \geq \gamma_a \cdot \frac{|C_a^\star|}{n} - \frac{\Lambda}{2}$$

If $i \in B_a(u)$, then, we have :

$$\mathbf{E}[\text{NCOUNT}(i, u)] = \mathbf{E}\left[\sum_{j \neq i, j \in T_u} \mathbf{1}\{N_i(u) = N_j(u)\}\right]$$

$$= \mathbf{E}\left[\sum_{j \neq i, j \in T_u \cap G_a(u)} \mathbf{1}\{N_i(u) = N_{\text{dom}_a}(u)\} + \sum_{j \neq i, j \in T_u \cap B_a(u)} \mathbf{1}\{N_i(u) = N_j(u)\}\right]$$

$$= \mathbf{E}\left[\sum_{j \neq i, j \in T_u \cap B_a(u)} \mathbf{1}\{N_i(u) = N_j(u)\}\right]$$

$$= \mathbf{E}[|T_u \cap B_a(u)|]$$

$$\leq \frac{|C_a^\star| - |S_a^{\text{dom}}|}{n} = \left(1 - \frac{|S_a^{\text{dom}}|}{|C_a^\star|}\right) \cdot \frac{|C_a^\star|}{n} = (1 - \gamma_a) \cdot \frac{|C_a^\star|}{n}$$

Using Hoeffding's inequality, with probability at least $1 - \exp\left(-\frac{\Lambda^2}{2|T_u|}\right)$

$$\text{NCOUNT}(i, u) < \mathbf{E}[\text{COUNT}(i, u)] + \frac{\Lambda}{2} < (1 - \gamma_a) \cdot \frac{|C_a^\star|}{n} + \frac{\Lambda}{2}$$

Set $\Lambda = \frac{|C_a^\star|}{n}(2\gamma_a - 1)$ and $|T_u| \geq \frac{4\log(nM/\delta)}{(2\gamma_a - 1)^2}$. Then, for any pair of entities $i, j \in C_a^\star$ such that $i \in G_a(u)$ and $j \in B_a(u)$, we have, with a probability $1 - \delta/nM^2$:

$$\text{NCOUNT}(i, u) > \text{NCOUNT}(j, u).$$

Using union bound for every pair of entities in $[M]$ and $u \in V$, with probability at least $1 - \delta$, we have the final claim. $\qquad\square$

From the previous Lemma C.3, we know that NCOUNT values are always larger for the dominant MAG partition, and therefore merging the neighborhoods of all the nodes gives us the dominant MAG. As dominant MAG is within a distance of at most $\beta \cdot n$ from every MAG in the cluster, the dominant MAG returned is a sufficiently good approximation of the true MAG. We formalize this using the following statement.

**Theorem C.4** (Theorem 3.4 Restated). *Suppose $\mathcal{M}_1, \mathcal{M}_2, \cdots \mathcal{M}_M$ satisfy $(\alpha, \beta)$ clustering property. If $\gamma_a > 1/2$ and $C_a^\star = \Omega(\frac{n \log(n/M\delta)}{(2\gamma_a - 1)^2})$ for all $a \in [k]$, then, Algorithm $(\alpha, \beta)$-RECOVERY recovers graphs $\widehat{\mathcal{M}}_1, \cdots \widehat{\mathcal{M}}_M$ such that for every entity $i \in [M]$, we have $d(\mathcal{M}_i, \widehat{\mathcal{M}}_i) \leq \beta n$ with probability $1 - \delta$. Moreover, every entity uses at most $(\Delta + 1) + \frac{4(\Delta+1)\log(M/\delta)}{(\alpha-\beta)^2}$ many atomic interventions.*

*Proof.* From Lemma C.3, we have that $\text{NCOUNT}(i, u) > \text{NCOUNT}(j, u)$, which implies $u_{\max} \in G_a(u)$. Using Algorithm 4, every entity $i$ in the cluster $C_a^\star$ is assigned the graph $\widehat{\mathcal{M}}_i = \mathcal{M}_a^{\text{dom}}$. From the definition of $(\alpha, \beta)$−clustering property, we have that all entities $i \in C_a^\star$ are such that $d(\mathcal{M}_i, \widehat{\mathcal{M}}_i) = d(\mathcal{M}_i, \mathcal{M}_a^{\text{dom}}) \leq \beta n$.

Using Algorithm 4 we assign every entity to a single node $u \in V$, and perform at most $\Delta + 1$ interventions to identify all the neighbors of $u$ for every entity in $T_u$. Therefore, we perform at most $\Delta + 1$ interventions per entity. For obtaining clusters, from Lemma 3.3, we know that every entity performs at most $\frac{4(\Delta+1)\log(M/\delta)}{(\alpha-\beta)^2}$ interventions. Hence, the theorem. $\square$

# D  Discovery under $\alpha$-Clustering Property

## D.1  From $\alpha$-Clustering to Learning Causal Graphs

Suppose that the underlying MAGs $\mathcal{M}_1, \mathcal{M}_2, \cdots, \mathcal{M}_M$ satisfy the $\alpha$-clustering property, our algorithms are based on first accurately recovering these clusters. The idea of going from clusters to MAGs is simple and is based on distributing the interventions across the entities in the cluster. We now discuss a meta-algorithm that returns the associated causal MAG of every entity given the true clustering. Our meta-algorithm takes as input the true clusters $C_1^\star, C_2^\star, \cdots, C_k^\star$ and recovers the MAGs associated with each of them. In any cluster $C_b^\star$ such that $|C_b^\star| < n$, our meta-algorithm uses an additional $\lceil n/|C_b^\star| \rceil$ many interventions for each entity in $C_b^\star$. For clusters satisfying $|C_b^\star| \geq n$, it uses an extra intervention per entity.

**Meta-Algorithm**. Consider a true cluster $C_b^\star$ ($b \in [k]$). Construct a mapping $\phi$ that partitions the $n$ nodes in $V$ among all the entities in $C_b^\star$, such that no entity is assigned to more than $\lceil n/|C_b^\star| \rceil$ many nodes. By definition, all entities in $C_b^\star$ have the same PAG. Let $\mathcal{U}$ be the common PAG. Construct a MAG $\mathcal{M}$ from $\mathcal{U}$ as follows. Consider an edge $(u, v)$ in $\mathcal{U}$. Let $u = \phi(i)$ and $v = \phi(j)$ where the entities $i, j \in C_b^\star$ are such that we intervene on node $u$ in entity $i$ and node $v$ in entity $j$ ($i$ could be equal to $j$). Now, if $v \in \text{ch}_i(u)$, we add $u \to v$ into the graph $\mathcal{M}$, else if $u \in \text{ch}_j(v)$, we add $u \leftarrow v$, and $u \leftrightarrow v$ otherwise. We assign graph $\mathcal{M}$ for every entity in $C_b^\star$. Repeating this procedure for every $C_b^\star$ generates the $M$ MAGs, one for each entity.

**Lemma D.1.** *Suppose there is an Algorithm $\mathcal{A}$ that recovers the true clusters $C_1^\star, C_2^\star, \cdots, C_k^\star$ of the underlying MAGs $\mathcal{M}_1, \mathcal{M}_2, \cdots, \mathcal{M}_M$ satisfying $\alpha$-clustering property such that every entity $i \in [M]$ uses at most $f(M)$ interventions. Then, there is an algorithm that can learn all the MAGs $\mathcal{M}_1, \mathcal{M}_2, \cdots, \mathcal{M}_M$ such that every entity $i \in [M]$ uses at most $f(M) + \lceil n/\Upsilon \rceil$ many interventions, where $\Upsilon = \min_{b \in [k]} C_b^\star$.*

*Proof.* Consider a cluster $C_b^\star$ for $b \in [k]$. As the mapping $\phi$ assigns every entity at most $\lceil n/|C_b^\star| \rceil$ many nodes to intervene on, we have that every entity in $C_b^\star$ uses at most $\lceil n/|C_b^\star| \rceil$ additional interventions. Therefore, over all true clusters, every entity uses at most $f(M) + \lceil n/\Upsilon \rceil$ many interventions.

Consider any cluster $C_b^\star$. The mapping $\phi$ in the Meta-Algorithm is well-defined and satisfies the claim that for every node $u \in V$, there exists an entity in $C_b^\star$ for which we construct an interventional distribution $\text{do}(u)$. Therefore, for cluster $C_b^\star$, we have $n$ interventional distributions one for every node in $V$, and we use Algorithm RECOVERG to learn the MAG for this cluster (i.e., MAG for all the entities in $C_b^\star$). Repeating this for every cluster $C_1^\star, C_2^\star, \cdots, C_k^\star$, we obtain all the MAGs $\mathcal{M}_1, \mathcal{M}_2, \cdots, \mathcal{M}_M$. $\square$

If the clusters are of size at least $n$, i.e., $\min_{b \in [k]} |C_b^\star| \geq n$, then, we have the following corollary from Lemma D.1.

**Corollary D.2.** *Suppose there is an Algorithm $\mathcal{A}$ that recovers the true clusters $C_1^\star, C_2^\star, \cdots, C_k^\star$ of the underlying MAGs $\mathcal{M}_1, \mathcal{M}_2, \cdots, \mathcal{M}_M$ satisfying the $\alpha$-clustering property such that every entity $i \in [M]$ uses at most $f(M)$ interventions. Suppose $\min_{b \in [k]} |C_b^\star| \geq n$. Then, there is an algorithm that can learn all the MAGs $\mathcal{M}_1, \mathcal{M}_2, \cdots, \mathcal{M}_M$ such that every entity $i \in [M]$ uses at most $f(M) + 1$ many interventions.*

## D.2 Discovery without Latents

In this section, we present a randomized algorithm that recovers (with high probability) all the $M$ MAGs $\mathcal{M}_1, \ldots, \mathcal{M}_M$ when the underlying data generating process for each of these entities do not have any latents (i.e., causal DAGs $\mathcal{D}_1, \ldots, \mathcal{D}_M$ satisfy causal sufficiency). This translates into the fact that the MAGs $\mathcal{M}_1, \ldots, \mathcal{M}_M$ do not have bidirected edges.

We first make the observation that to identify that two graphs, say $\mathcal{M}_i$ and $\mathcal{M}_j$ belong to different clusters, it suffices to find a node $u$ from the node-difference set $\operatorname{diff}(\mathcal{M}_i, \mathcal{M}_j)$ and checking their outgoing neighbors using Algorithm IDENTIFY-OUTNBR. We argue that, with probability at least $1 - \delta$, we can identify one such node $u \in \operatorname{diff}(\mathcal{M}_i, \mathcal{M}_j)$ by sampling $2 \log(M/\delta)/\alpha$ nodes uniformly from $V$ as $|\operatorname{diff}(\mathcal{M}_i, \mathcal{M}_j)| = d(\mathcal{M}_i, \mathcal{M}_j) \geq \alpha n$.

In Algorithm NOLATENTS, we obtain a sample of nodes $S$ and construct interventional distribution for every entity in $[M]$, and for every node in $S$. After finding the outgoing neighbors for every entity $i$ and node in $S$, we construct a graph $\mathcal{P}$ on entities (i.e., the node set of $\mathcal{P}$ is $[M]$). We include an edge between two entities if they share the same outgoing neighbors for every $u \in S$. This ensures that every entity is connected only to the entities belonging to the same true cluster, and we return the connected components in $\mathcal{P}$ as our clusters.

---

**Algorithm 5** NOLATENTS

1: **Input:** $\alpha > 0$, confidence parameter $\delta > 0$, PAGs $\mathcal{U}_1, \ldots, \mathcal{U}_M$ of $M$ entities.
2: **Output:** Partition of $[M]$ into clusters
3: Let $S$ denote a uniform sample of $\frac{2 \log M/\delta}{\alpha}$ nodes from $V$ selected with replacement.
4: **for** every entity $i \in [M]$ and $u \in S$ **do**
5: $\quad$ $\operatorname{ch}_i(u) \leftarrow$ IDENTIFY-OUTNBR$(i, u)$
6: **end for**
7: Let $\mathcal{P}$ denote an empty graph on set of entities $[M]$
8: **for** every pair of entities $i, j$ **do**
9: $\quad$ **if** $\operatorname{ch}_i(u) = \operatorname{ch}_j(u)$ and $\Gamma_i(u) = \Gamma_j(u)$ for every $u \in S$ **then**
10: $\quad\quad$ Add an edge between entities $i$ and $j$ in $\mathcal{P}$
11: $\quad$ **end if**
12: **end for**
13: Return connected components in $\mathcal{P}$

---

**Claim D.3.** *Let $S$ denote a set of $2 \log(M/\delta)/\alpha$ nodes sampled with replacement uniformly from $V$. Then, for every pair of entities $i, j$ that belong to different true clusters, we have with probability at least $1 - \delta$, $u \in \operatorname{diff}(\mathcal{M}_i, \mathcal{M}_j)$ for some $u \in S$.*

*Proof.* Let $S$ denote a set of sampled nodes such that $|S| = 2 \log(M/\delta)/\alpha$. Therefore, we have
$$\Pr_{u \sim V}[u \in \operatorname{diff}(\mathcal{M}_i, \mathcal{M}_j)] \geq \alpha, \text{ and}$$
$$\Pr_{S \sim V}[\forall u \in S : u \notin \operatorname{diff}(\mathcal{M}_i, \mathcal{M}_j)] \leq (1 - \alpha)^{|S|}$$
$$\leq e^{-\alpha |S|} \leq \frac{\delta}{M^2}.$$

Using union bound for every pair of entities in $[M]$ that belong to two different clusters, we have:
$$\forall i, j \in [M], \ \Pr_{S \sim V}[\forall u \in S, u \notin \operatorname{diff}(\mathcal{M}_i, \mathcal{M}_j)] \leq \delta.$$

Therefore, for every pair of entities $i, j \in [M]$ belonging to different true clusters, there exists $u \in S$ such that :
$$\Pr[u \in \operatorname{diff}(\mathcal{M}_i, \mathcal{M}_j)] \geq 1 - \delta.$$

$\square$

**Lemma D.4.** *Assume causal sufficiency. If MAGs $\mathcal{M}_1, \ldots, \mathcal{M}_M$ satisfy $\alpha$-clustering property with true clusters $C_1^\star, \ldots, C_k^\star$, then Algorithm NOLATENTS exactly recovers the clusters $C_1^\star, \ldots, C_k^\star$ with probability at least $1 - \delta$. Every entity $i \in [M]$ uses $2\log(M/\delta)/\alpha$ many atomic interventions.*

*Proof.* Consider two entities $i, j$ and their corresponding MAGs $\mathcal{M}_i$ and $\mathcal{M}_j$ respectively. We first observe that if the PAGs of these two entities are different then they belong to different clusters. Now consider the case where the PAGs for both these entities are the same, i.e., $\mathcal{U}_i = \mathcal{U}_j$. Now if $i$ and $j$ belong to different true clusters, then we claim that it suffices to find a node $u$ from the node-difference set $\mathrm{diff}(\mathcal{M}_i, \mathcal{M}_j) = \{u \mid N_i(u) \neq N_j(u)\}$ to notice this fact. As there are no latents (causal sufficiency), we can identify whether $u \in \mathrm{diff}(\mathcal{M}_i, \mathcal{M}_j)$, by checking only the outgoing neighbors of $u$ for entities $i, j$, i.e., $\mathrm{diff}(\mathcal{M}_i, \mathcal{M}_j) = \{u \mid \mathrm{ch}_i(u) \neq \mathrm{ch}_j(u)\}$. When we identify such a node $u$, the set of outgoing neighbors of node $u$ are different for entities $i, j$, and therefore must belong to different true clusters (by $\alpha$-clustering property). In order to identify at least one node $u \in \mathrm{diff}(\mathcal{M}_i, \mathcal{M}_j)$, we use sampling.

Let $S$ denote the set of sampled nodes (with replacement) from $V$ such that $|S| = 2\log(M/\delta)/\alpha$. In Algorithm NOLATENTS, we construct interventional distributions for every node $u \in S$, for every entity $i \in [M]$. Using these interventional distributions we obtain the outgoing neighbors of nodes in $S$ using Algorithm IDENTIFY-OUTNBR.

From Claim D.3, we have that for every pair of entities $i, j$ belonging to different true clusters, there exists $u \in S$ such that:

$$\Pr[\mathrm{ch}_i(u) \neq \mathrm{ch}_j(u)] = \Pr[u \in \mathrm{diff}(\mathcal{M}_i, \mathcal{M}_j)] \geq 1 - \delta.$$

This implies that, with probability at least $1 - \delta$, for every $i, j$ pair we have the following: in the entity graph $\mathcal{P}$ there would not be an edge between $i, j$ if they belong to different true clusters, and there would be an edge if they belong to the same true cluster. The resulting graph $\mathcal{P}$, will have $k$ connected components and Algorithm NOLATENTS will return the true clusters correctly.

Hence, with probability at least $1 - \delta$, we can recover all the true clusters $C_1^\star, \ldots, C_k^\star$ using Algorithm NOLATENTS. $\square$

**Theorem D.5.** *Assume causal sufficiency. If MAGs $\mathcal{M}_1, \ldots, \mathcal{M}_M$ satisfy $\alpha$-clustering property with true clusters $C_1^\star, \ldots, C_k^\star$ then Algorithm NOLATENTS exactly recovers these clusters with probability at least $1 - \delta$. Furthermore, if $\min_{b \in [k]} |C_b^\star| \geq n$, then there is an algorithm that exactly learns all these MAGs with probability at least $1 - \delta$. Every entity $i \in [M]$ uses $2\log(M/\delta)/\alpha + 1$ many atomic interventions.*

*Proof.* From Lemma D.4, we can recover the clusters correctly with probability at least $1 - \delta$. Using the Meta-Algorithm discussed in Section D.1, we can learn the graphs of every entity with a single additional intervention (see Corollary D.2). This establishes the result. $\square$

## D.3 Discovery with Latents: Bounded Degree MAGs

Throughout this section, we let :

$$\Delta = \max_{i \in [M], u \in V} |\Gamma_i(u)|.$$

We now discuss an algorithm that recovers clusters $C_1^\star, C_2^\star \cdots, C_k^\star$ using ideas developed in Section D.2 but now with latents in the system. In the presence of latents, the collection of MAGs $\mathcal{M}_1, \ldots, \mathcal{M}_M$ are mixed graphs that also contain bidirected edges, which introduces issues, as bidirected edges cannot be detected easily. For example, two entities $i$ and $j$ might be such that $u \leftrightarrow v$ could be present in $\mathcal{M}_i$ and $u \leftarrow v$ could be present in $\mathcal{M}_j$, in which case intervening on just $u$ alone will not suffice to distinguish $i$ from $j$, we need interventions on both $u$ and $v$. This is the idea behind Algorithm $\alpha$-BOUNDEDDEGREE, which identifies all the outgoing and bidirected edges incident on the set of sampled nodes (say $S$), for every entity in $[M]$. Since from this we can compute all neighboring relations of $u$ ($N_i(u)$), Algorithm $\alpha$-BOUNDEDDEGREE then checks whether these neighborhoods are the same or not for every node $u \in S$. We can now leverage the $\alpha$-clustering property to argue that this process succeeds with probability at least $1 - \delta$.

As we use Algorithm IDENTIFY-BIDIRECTED, to find all bidirected edges incident on a node $u \in S$, we use an additional $O(\Delta)$ atomic interventions (per entity) where $\Delta = \max_{i \in [M], u \in V} \Gamma_i(u)$ is the maximum undirected degree in the PAGs $\mathcal{U}_1, \ldots, \mathcal{U}_M$.

---

**Algorithm 6** $\alpha$-BOUNDEDDEGREE

1: **Input:** $\alpha > 0$, confidence parameter $\delta > 0$, PAGs $\mathcal{U}_1, \ldots, \mathcal{U}_M$ of $M$ entities
2: **Output:** Partition of $[M]$ into clusters
3: Let $S$ denote a uniform sample of $\frac{2 \log M/\delta}{\alpha}$ nodes from $V$ selected with replacement.
4: **for** every entity $i \in [M]$ and $u \in S$ **do**
5:     $\text{ch}_i(u) \leftarrow$ IDENTIFY-OUTNBR$(i, u)$
6:     $\text{sp}_i(u) \leftarrow$ IDENTIFY-BIDIRECTED$(i, u)$
7:     $\text{pa}_i(u) \leftarrow \Gamma_i(u) \setminus (\text{ch}_i(u) \cup \text{sp}_i(u))$
8:     Construct $N_i(u)$ (defined in (2))
9: **end for**
10: Let $\mathcal{P}$ denote an empty graph on set of entities $[M]$
11: **for** every pair of entities $i, j$ **do**
12:     **if** $N_i(u) = N_j(u)$ for every $u \in S$ **then**
13:         Include an edge between $i$ and $j$ in $\mathcal{P}$
14:     **end if**
15: **end for**
16: Return connected components in $\mathcal{P}$

---

**Lemma D.6.** *If the underlying MAGs $\mathcal{M}_1, \ldots, \mathcal{M}_M$ satisfy $\alpha$-clustering property with true clusters $C_1^\star, \ldots, C_k^\star$, then Algorithm $\alpha$-BOUNDEDDEGREE exactly recovers the clusters $C_1^\star, \ldots, C_k^\star$ with probability at least $1 - \delta$. Every entity $i \in [M]$ uses at most $2(\Delta + 1) \log(M/\delta)/\alpha$ many atomic interventions.*

*Proof.* We follow a proof idea similar to Lemma D.4. Again if two entities $i, j$ have different PAGs then they belong to different true clusters.

Consider two entities $i, j$ belonging to different true clusters but having the same PAG. Again it suffices to find a node $u$ from the node-difference set $\text{diff}(\mathcal{M}_i, \mathcal{M}_j) = \{u \mid N_i(u) \neq N_j(u)\}$ to conclude that they belong to different clusters.

As there are latents (causal sufficiency), we cannot identify whether $u \in \text{diff}(\mathcal{M}_i, \mathcal{M}_j)$, by checking only the outgoing neighbors of $u$ for entities $i, j$, and have to check the set of bidirected edges incident on $u$ as well. We can identify all the bidirected edges incident on $u$ for both $i, j$ using Algorithm IDENTIFY-BIDIRECTED. Identifying such a node $u$, whose set of neighbors of node $u$ are different for entities $i, j$, provides a certificate that $i, j$ belong to different true clusters ($\alpha$-clustering property). In order to identify at least one node $u \in \text{diff}(\mathcal{M}_i, \mathcal{M}_j)$, we use sampling.

Let $S$ denote the set of sampled nodes (with replacement) from $V$ such that $|S| = 2 \log(M/\delta)/\alpha$. In Algorithm $\alpha$-BOUNDEDDEGREE, we construct interventional distributions for every node $u \in S$ and all the neighbors in the PAG given by $\Gamma_i(u)$, for every entity $i \in [M]$. From these interventional distributions, we can compute $N_i(u)$ and $N_j(u)$ for all the nodes $u \in S$ (using Algorithms IDENTIFY-OUTNBR and IDENTIFY-BIDIRECTED).

From Claim D.3, we have that for every pair of entities $i, j$ belonging to different true clusters, there exists $u \in S$ such that:

$$\Pr[N_i(u) \neq N_j(u)] = \Pr[u \in \text{diff}(\mathcal{M}_i, \mathcal{M}_j)] \geq 1 - \delta.$$

Hence, with probability at least $1 - \delta$, we can recover all the true clusters using Algorithm $\alpha$-BOUNDEDDEGREE.

For an entity $i$, the total number of interventional distributions constructed is

$$\sum_{u \in S} (1 + |\Gamma_i(u)|) \leq |S|(\Delta + 1) = 2(\Delta + 1) \log(M/\delta)/\alpha \ \text{ as } \ \max_{i \in [M], w \in V} |\Gamma_i(w)| \leq \Delta.$$

$\square$

**Theorem D.7.** *If the underlying MAGs $\mathcal{M}_1, \ldots, \mathcal{M}_M$ satisfy $\alpha$-clustering property with true clusters $C_1^\star, \ldots, C_k^\star$, then Algorithm $\alpha$-BOUNDEDDEGREE exactly recovers these clusters with probability at least $1 - \delta$. Furthermore, if $\min_{b \in [k]} |C_b^\star| \geq n$, then there is an algorithm that exactly learns all these MAGs with probability at least $1 - \delta$. Every entity $i \in [M]$ uses at most $2(\Delta + 1) \log(M/\delta)/\alpha + 1$ many atomic interventions.*

*Proof.* From Lemma D.6, we can recover the clusters correctly with probability at least $1 - \delta$. Using the Meta-Algorithm discussed in Section D.1, and from Corollary D.2, we can obtain an algorithm to learn the graphs of every entity with an additional intervention per entity. This completes the proof. $\qquad\square$

## D.4  Missing Details from Section 4

In Algorithm $\alpha$-GENERAL, we obtain all the outgoing neighbors of the sampled set of nodes $S$. Then, we construct a graph on set of entities, $[M]$ such that an edge between a pair of entities $i, j$ is included if they share same PAGs, i.e., $\mathcal{U}_i = \mathcal{U}_j$ and same outgoing neighbors for every node in $S$. However, it is possible that the graph $\mathcal{P}$ can contain more than one true cluster. In the next lemma, we show that we can detect this, and remove all the edges between entities belonging to two different clusters using 2 interventions.

**Lemma D.8.** *Suppose a component $T_a$ in $\mathcal{P}_{\text{itr}}$ for some $\text{itr} \geq 1$ contains all the entities from two true clusters $C_b^\star, C_c^\star$. If $\min_{r \in [k]} |C_r^\star| \geq \Omega(n \log M/\delta)$, then, we can identify, with a probability $1 - \delta/2k^2$, all the pairs of entities $i', j' \in T_a$ such that $i' \in C_b^\star$ and $j' \in C_c^\star$ (or vice-versa) using at most 2 interventions for every entity in $T_a$.*

*Proof.* We claim that if a component $T_a$ containing $C_b^\star$ and $C_c^\star$ exists, then, we can identify a pair of entities $i, j$ that are joined by an edge in $\mathcal{P}_{\text{itr}}$ such that $i \in C_b^\star$ and $j \in C_c^\star$ or vice-versa.

It suffices to find a node $u$ from the node-difference set $\text{diff}(\mathcal{M}_i, \mathcal{M}_j) = \{u \mid N_i(u) \neq N_j(u)\}$ to conclude that they belong to different clusters. From Claim D.3, we know that when $|S| = 2\log(2M/\delta)/\alpha$, we can identify such a $u \in \text{diff}(\mathcal{M}_i, \mathcal{M}_j)$ with a probability $1 - \delta/2$. We make the observation that a pair of entities $i, j$ that have an edge in this $\mathcal{P}_{\text{itr}}$ and from different true clusters, can differ only if there is a node $u \in \text{diff}(\mathcal{M}_i, \mathcal{M}_j)$ such that $u$ has a bidirected edge $u \leftrightarrow v$ in $\mathcal{M}_i$, and a directed edge $u \leftarrow v$ in $\mathcal{M}_j$ (or vice-versa). Intervening on both $u$ and $v$ will separate these entities, our main idea is to ensure that this happens.

Consider a mapping $\pi : [M] \rightarrow V$ where $\pi(i)$ is assigned a node from $V$ selected uniformly at random. Using this mapping, we ensure that there are two entities $i \in T_a \cap C_b^\star, j \in T_a \cap C_c^\star$ joined by an edge, such that $\pi(i) = \pi(j) = v$ and $u \leftrightarrow v$ in $\mathcal{M}_i$, $u \leftarrow v$ in $\mathcal{M}_j$ (or vice-versa) for some $u \in S$. We have:

$$\Pr[\text{ for any } i \in T_a, \ \pi(i) \neq v] = 1 - \frac{1}{n}, \text{ and}$$

$$\Pr[\forall i \in C_b^\star \ : \pi(i) \neq v] = \left(1 - \frac{1}{n}\right)^{|C_b^\star|}.$$

Similarly, we have

$$\Pr[\forall j \in C_c^\star \ : \pi(j) \neq v] = \left(1 - \frac{1}{n}\right)^{|C_c^\star|}.$$

$$\Pr[\forall \ i \in C_b^\star, j \in C_c^\star \text{ such that } \pi(i) \neq v, \ \pi(j) \neq v] = \left(1 - \frac{1}{n}\right)^{|C_b^\star| + |C_c^\star|}$$

$$\leq \frac{\delta}{2M^2} \leq \frac{\delta}{2k^2}$$

$$\Rightarrow \Pr[\exists i \in C_b^\star, \exists j \in C_c^\star \ : \pi(i) = \pi(j) = v] \geq 1 - \frac{\delta}{2k^2}.$$

As we intervene on $\pi(i)$ for every entity $i \in T_a$, we know that there exists $i \in C_b^\star, j \in C_c^\star$, both in $T_a$ and that are assigned $v$ by $\pi$. Therefore, we can separate $i, j$ and remove the edge from $\mathcal{P}_{\text{itr}}$. Now, we create an intervention on $\pi(i) = \pi(j) = v$ for every entity in $T_a$ and separate all the entity pairs

$(i', j')$ joined by an edge in $\mathcal{P}_{\text{itr}}$ that satisfy: $u \leftarrow v$ in $\mathcal{M}_{i'}$ and $u \leftrightarrow v$ in $\mathcal{M}_{j'}$ (or vice-versa). As we use at most two interventions for every entity in $T_a$, the lemma follows. $\qquad\square$

---

**Algorithm 7** $\alpha$-GENERAL

---

**Input:** $\alpha > 0$, confidence parameter $\delta > 0$, PAGs $\mathcal{U}_1, \ldots, \mathcal{U}_M$ of $M$ entities $\mathcal{U}_1, \ldots, \mathcal{U}_M$)
**Output:** Partition of $[M]$ into clusters
Let $S$ denote a uniform sample of $\frac{2 \log(2M/\delta)}{\alpha}$ nodes from $V$ selected with replacement.
**for** every entity $i \in [M]$ and $u \in S$ **do**
   $\text{ch}_i(u) \leftarrow \text{IDENTIFY-OUTNBR}(i, u)$
**end for**
Let $\mathcal{P}$ denote an empty graph on the set of entities $[M]$.
**for** every pair of entities $i, j$ **do**
   **if** $\text{ch}_i(u) = \text{ch}_j(u)$ and $\Gamma_i(u) = \Gamma_j(u)$ $\forall u \in S$ **then**
      Include an edge between $i$ and $j$ in $\mathcal{P}$
   **end if**
**end for**
$\text{itr} \leftarrow 1, \mathcal{P}_0 \leftarrow \mathcal{P}$
**while** TRUE **do**
   $\mathcal{P}_{\text{itr}} \leftarrow \mathcal{P}_{\text{itr}-1}$
   Let $T_1, T_2, \cdots$ denote the components in $\mathcal{P}_{\text{itr}}$.
   For all $i \in [M]$, obtain interventional distribution on $\pi(i)$ picked u.a.r from $V$.
   **if** $\exists$ edge $(i, j) \in \mathcal{P}_{\text{itr}}$ in component $T_a$ such that $\pi(i) = \pi(j)$ **then**
      Let $v = \pi(i) = \pi(j)$
      **if** $v \in \text{sp}_i(u), v \notin \text{sp}_j(u)$ (or vice-versa) for some $u \in S$ **then**
         Intervene on $v$ for every entity in $T_a$.
         Remove edge $(i', j')$ from $\mathcal{P}_{\text{itr}}$ if $v \in \text{sp}_{i'}(u)$, $v \notin \text{sp}_{j'}(u)$ (or vice-versa) for every $i', j' \in T_a$
      **end if**
   **end if**
   **if** the set of edges in $\mathcal{P}_{\text{itr}}$ are *same* as the set of edges in $\mathcal{P}_{\text{itr}-1}$ **then**
      Return connected components in $\mathcal{P}_{\text{itr}}$
   **end if**
   $\text{itr} \leftarrow \text{itr} + 1$
**end while**

---

**Lemma D.9.** *If the underlying MAGs satisfy $\alpha$-clustering property with true clusters $C_1^\star, \ldots, C_k^\star$ such that $\min_{b \in [k]} C_b^\star = \Omega(n \log(M/\delta))$ entities, the Algorithm $\alpha$-GENERAL exactly recovers the clusters $C_1^\star, \ldots, C_k^\star$ with probability at least $1 - \delta$. Every entity $i \in [M]$ uses at most $O(\log(M/\delta)/\alpha + k^2)$ many atomic interventions.*

*Proof.* From Claim D.3, with probability at least $1 - \delta/2$, we have that the set of sampled nodes $S$ (where $|S| = 2\log(2M/\delta)/\alpha$) satisfy that for every pair of entities from different clusters there is a node $u \in S$ that can be used to identify that they belong to different clusters. Using Lemma D.8, we have that, in every iteration itr, we remove all the edges in $\mathcal{P}_{\text{itr}}$ between entities that are part of the same component but from different true clusters. After $k^2$ iterations, we would have separated all the pairs of entities between all the true clusters. In Algorithm $\alpha$-GENERAL, we return the connected components in $\mathcal{P}_{\text{itr}}$ when there is no change in the set of edges between entities between $\mathcal{P}_{\text{itr}-1}$ and $\mathcal{P}_{\text{itr}}$.

From Lemma D.8, we have that, every entity performs at most $|S| + 2k^2$ interventions. As there are at most $k^2$ iterations, and from Lemma D.8, each iteration fails with probability at most $\delta/2k^2$, using union bound, we have that at least one of the iterations fails with probability at most $\delta/2$.

Finally, using union bound for failure probability of calculating $S$ correctly, and failing in at least one of the iterations, we have, with probability at least $1 - \delta$, Algorithm $\alpha$-GENERAL recovers the true clusters. $\qquad\square$

From Lemma D.9, we know that we can recover the clusters correctly with probability at least $1 - \delta$. Using the Meta-Algorithm discussed in Appendix D.1, and from Corollary D.2, we can obtain an algorithm to learn the graphs of every entity with an additional intervention per entity. Combining it with guarantees obtained by Algorithm $\alpha$-BOUNDEDDEGREE in Theorem D.7, gives us the following result.

**Theorem D.10** (Theorem 4.1 Restated). *If MAGs $\mathcal{M}_1, \ldots, \mathcal{M}_M$ satisfy $\alpha$-clustering property with true clusters $C_1^\star, \ldots, C_k^\star$ such that $\min_{b \in [k]} |C_b^\star| = \Omega(n \log(M/\delta))$. Then, there is an algorithm that exactly learns all these MAGs with probability at least $1 - \delta$. Every entity $i \in [M]$ uses $\min \left\{ O(\Delta \log(M/\delta)/\alpha), O(\log(M/\delta)/\alpha + k^2) \right\}$ many atomic interventions.*

### D.5 Lower Bound on the Number of Interventions

In this section, we present a lower bound for the number of interventions required by every entity to recover true clusters. First, we state Yao's minimax theorem, which will be used to prove the lower bound.

**Theorem D.11** (Yao's minimax theorem [Yao, 1977]). *Let $\mathcal{X}$ be a set of inputs to a problem and $\mathcal{A}$ the set of all possible deterministic algorithms that solve the problem. For any algorithm $A \in \mathcal{A}$ and $x \in \mathcal{X}$, let $cost(A, x)$ denote real-valued measure of cost of an algorithm $A$ on input $x$. Let $\nu, \mu$ be distributions over $\mathcal{A}$ and $\mathcal{X}$ respectively. Then,*

$$\max_{x \in \mathcal{X}} \mathop{\mathbf{E}}_{A \sim \nu \mathcal{A}} [cost(A, x)] \geq \min_{a \in \mathcal{A}} \mathop{\mathbf{E}}_{X \sim \mu \mathcal{X}} [cost(a, X)]$$

Informally, the theorem states that to prove lower bounds on the cost of *any* randomized algorithm, we have to find some distribution $\mu$ on inputs, such that *every* deterministic algorithm $A \in \mathcal{A}$ has high cost.

**Outline of the Lower Bound.** Our distribution $\mu$ places a probability of $1/2$ for pairs of MAGs that have distance zero and a probability of $1/2$ equally distributed among all pairs of MAGs with distance equal to $\alpha n$. This ensures that both the events considered are equally likely, and we show that to distinguish them, with success probability at least $2/3$ (over the distribution $\mu$), every deterministic algorithm must make $\Omega(1/\alpha)$ interventions for both the MAGs. Then, we use Yao's theorem to translate this into a worst case lower bound for any randomized algorithm. In particular, this means that any algorithm that is based on recovering the clusters to construct the MAGs will require $\Omega(1/\alpha)$ interventions for every entity in $[M]$.

**Details.** For the lower bound, consider the case when $M = 2$, and assuming causal sufficiency, where we wish to identify the clusters of two MAGs $\mathcal{M}_1, \mathcal{M}_2$. We observe that a lower bound on the number of interventions required for every entity in the case of identifying two clusters will also extend for the general case of identifying $k$ clusters with latents.

Consider two MAGs $\mathcal{M}_1, \mathcal{M}_2$ on a node set $V$, with the promise that either $d(\mathcal{M}_1, \mathcal{M}_2) = 0$ or $d(\mathcal{M}_1, \mathcal{M}_2) = \alpha n$, and the goal is to identify which case holds. Note that in the first case the two entities are in the same cluster ($k = 1$), and in the second case they are in different clusters ($k = 2$).

Let $V = \{v_1, \ldots, v_n\}$ be the set of observable nodes of these MAGs. Consider the node difference set of the MAGs $\mathcal{M}_1, \mathcal{M}_2$ given by $\text{diff}(\mathcal{M}_1, \mathcal{M}_2)$ and let $e \in \{0, 1\}^n$ denote its characteristic vector where $l$th coordinate of $e$ is 1 iff $v_l \in \text{diff}(\mathcal{M}_1, \mathcal{M}_2)$. We can observe that, under the above promise, $e$ is either $0^n$ or has exactly $\alpha n$ ones. Therefore, we have reduced our problem to that of finding whether the vector $e$ contains all zeros or not. Using this reduction, we focus on establishing a lower bound for this modified problem.

We want to check if a given $n$-dimensional binary vector is a zero vector, i.e., $0^n$ or not, with a promise that if it is not a zero vector, then, it contains $\alpha n$ coordinates with 1 in them. Using Lemma D.12, we show that $\Omega\left(\frac{1}{\alpha}\right)$ queries to co-ordinates of $x$ are required, for any randomized or deterministic algorithm to distinguish between these two cases.

**Lemma D.12.** *Suppose we are given a vector $x \in \{0, 1\}^n$ with the promise that either $x = 0^n$ or $x$ contains $\alpha n$ ones. In order to distinguish these two cases with probability more than $2/3$, every randomized or deterministic algorithm must make at least $\Omega(1/\alpha)$ queries to the coordinates of the vector $x$.*

*Proof.* It is easy to see that every deterministic algorithm for this problem requires $(1-\alpha)n + 1$ queries. For obtaining a lower bound on the number of queries of any randomized algorithm, we use Yao's minimax theorem Yao [1977]. To do so, we construct an input distribution $\mu$ on $\{0,1\}^n$ and show that every deterministic algorithm on the worst case requires at least $q$ queries while succeeding with a probability $2/3$. From Yao's minimax theorem (Thm D.11), this implies that every randomized algorithm requires at least $q$ queries to output the correct answer with probability of success $2/3$. We construct $\mu$ by using a probability of $1/2$ for $0^n$ vector and a probability of $1/2$ equally distributed among all vectors in $\{0,1\}^n$ containing exactly $\alpha n$ ones.

Suppose a deterministic algorithm (denoted by ALG) is used to identify whether $x = 0^n$ or not. Let $\mathcal{E}(x)$ denote the event that the ALG answers correctly on $x \in \{0,1\}^n$, $Q(x)$ denote the set of queries used by ALG such that $|Q(x)| = q$ and $L(x)$ denote the coordinates of $x$ that are non-zero.

Consider the event $\mathcal{E}(x)$ when ALG answers correctly. We can write it as :

$$\Pr_{x \sim \mu}[\mathcal{E}(x)]$$
$$= \Pr[\mathcal{E}(x) \mid Q(x) \cap L(x) \neq \phi]\Pr[Q(x) \cap L(x) \neq \phi] + \Pr[\mathcal{E}(x) \mid Q(x) \cap L(x) = \phi]\Pr[Q(x) \cap L(x) = \phi].$$

We calculate the probability that the coordinates queried are not part of the non-zero coordinates of $x$, given by $Q(x) \cap L(x) = \phi$ :

$$\Pr_{x \sim \mu}[Q(x) \cap L(x) = \phi]$$
$$= \Pr[Q(x) \cap L(x) = \phi \mid x = 0^n]\Pr[x = 0^n] + \Pr[Q(x) \cap L(x) = \phi \mid x \neq 0^n]\Pr[x \neq 0^n]$$
$$= \frac{1}{2}\left(\Pr[Q(x) \cap L(x) = \phi \mid x = 0^n] + \Pr[Q(x) \cap L(x) = \phi \mid x \neq 0^n]\right)$$
$$= \frac{1}{2}\left(1 + \frac{\binom{n-q}{\alpha n}}{\binom{n}{\alpha n}}\right) = \frac{1 + \tau}{2} \quad \text{, where } \tau = \frac{\binom{n-q}{\alpha n}}{\binom{n}{\alpha n}}.$$

Now, we calculate the probability that ALG answers correctly when the queries $Q(x)$ all return zero. We upper bound this probability by considering the case when ALG answers 'yes', and the case when ALG answers 'no' separately. It is easy to observe that $\mathcal{E}(x)$ is correct when ALG ='yes' iff $x = 0^n$. Therefore, we have:

$$\Pr[\mathcal{E}(x) \mid Q(x) \cap L(x) = \phi] \leq \max \left\{ \overbrace{\frac{\Pr[\mathcal{E}(x), Q(x) \cap L(x) = \phi]}{\Pr[Q(x) \cap L(x) = \phi]}}^{\text{ALG answers 'yes'}}, \overbrace{\frac{\Pr[\mathcal{E}(x), Q(x) \cap L(x) = \phi]}{\Pr[Q(x) \cap L(x) = \phi]}}^{\text{ALG answers 'no'}} \right\}$$

$$\leq \max\left\{\frac{1/2}{(1+\tau)/2}, \frac{\tau/2}{(1+\tau)/2}\right\} \leq \frac{1}{1+\tau}.$$

$$\Pr_{x \sim \mu}[\mathcal{E}(x)] = \Pr[\mathcal{E}(x) \mid Q(x) \cap L(x) \neq \phi]\Pr[Q(x) \cap L(x) \neq \phi] + \Pr[\mathcal{E}(x) \mid Q(x) \cap L(x) = \phi]\Pr[Q(x) \cap L(x) = \phi]$$
$$\leq \Pr[Q(x) \cap L(x) \neq \phi] + \Pr[\mathcal{E}(x) \mid Q(x) \cap L(x) = \phi]\Pr[Q(x) \cap L(x) = \phi]$$
$$\leq \left(1 - \frac{1+\tau}{2}\right) + \frac{1}{1+\tau}\frac{1+\tau}{2} = 1 - \frac{\tau}{2}.$$

We know the probability of success for ALG is at least $\frac{2}{3}$. Therefore, we have $\Pr_{x \sim \mu}[\mathcal{E}(x)] \geq \frac{2}{3}$, which implies $\tau \leq \frac{2}{3}$.

Let $H(x)$ denote the binary entropy function. Using the bound from (MacWilliams and Sloane [1977], Page 309)

$$\sqrt{\frac{a}{8b(a-b)}}2^{aH(b/a)} \leq \binom{a}{b} \leq \sqrt{\frac{a}{2\pi b(a-b)}}2^{aH(b/a)}.$$

We have

$$\tau = \frac{\binom{n-q}{\alpha n}}{\binom{n}{\alpha n}} \leq \sqrt{\frac{8(n-q)(n-\alpha n)}{2\pi n(n-q-\alpha n)}} 2^{(n-q)H(\frac{\alpha n}{n-q})-nH(\alpha)}$$

$$= \sqrt{\frac{4}{\pi}\left(1 + \frac{q\alpha}{n-q-\alpha n}\right)} 2^{(n-q)(H(\frac{\alpha n}{n-q})-H(\alpha))-qH(\alpha)}.$$

We observe that $q \leq (1-\alpha)n + 1$ for any algorithm, as we can identify whether $x = 0^n$ or not trivially by querying more than $(1-\alpha)n + 1$ coordinates. Therefore,

$$\tau \leq \sqrt{\frac{4}{\pi}(1-q\alpha)} 2^{(n-q)(H(\frac{\alpha n}{n-q})-H(\alpha))-qH(\alpha)}$$

$$\leq \sqrt{\frac{4}{\pi}} 2^{-q\alpha \log e/2 + (n-q)(H(\frac{\alpha n}{n-q})-H(\alpha))-qH(\alpha)}$$

Using $\frac{\alpha n}{n-q} \geq \alpha$ and mean-value theorem, we have:

$$(n-q)\left(H\left(\frac{\alpha n}{n-q}\right) - H(\alpha)\right) \leq q\alpha H'\left(\frac{\alpha n}{n-q}\right)$$

$$= q\alpha \log\left(\frac{n-q}{\alpha n} - 1\right)$$

$$\leq q\alpha \log(1-\alpha) - q\alpha \log \alpha.$$

Substituting the above expression and expanding $H(\alpha)$, we have :

$$\tau \leq \sqrt{\frac{4}{\pi}} 2^{-q\alpha \log e/2 + q\alpha \log(1-\alpha) - q\alpha \log \alpha + q\alpha \log \alpha + (1-\alpha)q \log(1-\alpha)}$$

$$\leq \sqrt{\frac{4}{\pi}} 2^{-q\alpha \log e/2 + q \log(1-\alpha)}$$

$$\leq \sqrt{\frac{4}{\pi}} 2^{-q\alpha \log e/2 - q\alpha} \leq \frac{2}{3}.$$

Therefore, for ALG to succeed with probability at least $2/3$, we have

$$q \geq \Omega\left(\frac{1}{\alpha}\right).$$

Using this with Yao's minimax theorem (Thm D.11), we get that with every randomized algorithm needs $\Omega(1/\alpha)$ queries to succeed on this problem with probability at least $2/3$. $\qquad\square$

For the above problem of identifying whether a vector is zero or not, we can replace each coordinate query by an intervention on the corresponding node for the two entities (due to the equivalency between the two as explained above). Therefore, from Lemma D.12, we have the following corollary about recovering the clusters.

**Corollary D.13.** *Suppose we are given two MAGs $\mathcal{M}_1$ and $\mathcal{M}_2$ corresponding to two entities, with the promise that either $d(\mathcal{M}_1, \mathcal{M}_2) = 0$ or $d(\mathcal{M}_1, \mathcal{M}_2) = \alpha n$. In order to distinguish these two cases with probability at least $2/3$, every (randomized or deterministic) algorithm must make at least $\Omega(1/\alpha)$ interventions on both the entities.*

**Theorem D.14** (Theorem 4.2 Restated)**.** *Suppose the underlying MAGs $\mathcal{M}_1, \ldots, \mathcal{M}_M$ satisfy $\alpha$-clustering property. In order to recover the clusters with probability $2/3$, every (randomized or deterministic) algorithm requires $\Omega(1/\alpha)$ interventions for every entity in $[M]$.*

*Proof.* From Corollary D.13, we have that to identify whether two MAGs belong to the same cluster or not, we have to make at least $\Omega(1/\alpha)$ interventions for every entity. Therefore, to recover all the clusters, we have to make at least $\Omega(1/\alpha)$ many interventions for every entity $i \in [M]$. $\qquad\square$

**Note**. The lower bound for the number of interventions per entity is for the first step of identifying the underlying clustering. Similar to our upper bounds, our lower bound considers the worst-case, where the MAGs satisfy the $\alpha$-clustering property are all Markov equivalent. This implies that the PAGs obtained using FCI are identical and will not be helpful in identifying the clusters. In practice, the information available in the PAGs could be useful to reduce the number of interventions.

## E   Experimental Evaluation

In this section, we provide additional details about the experimental evaluation discussed in Section 5. For ensuring reproducibility, the entire experiment setup is submitted as part of the supplementary material.

### E.1   Learning MAGs under $(\alpha, \beta)$-clustering property

**(Synthetic) Data Generation**. We use following process for each of the five considered causal network (*Asia*, *Earthquake*, *Sachs*, *Survey*, and *Erdős-Renyi*). We construct causal MAGs for $M$ entities distributed among the clusters $C_1^\star, C_2^\star, \cdots C_k^\star$ equally, i.e., $|C_i^\star| = M/k$ for all $i \in [k]$. In our experiments, we set $k = 2$ (i.e., two clusters), and start with $k = 2$ DAGs that are sufficiently far apart. To do so, we create two copies of the original causal network $\mathcal{D}$, and denote the DAG copies by $\mathcal{D}_1$ and $\mathcal{D}_2$. For each of the DAGs $\mathcal{D}_1$ and $\mathcal{D}_2$, we select a certain number of pairs of nodes randomly, and include a latent variable between them, that has a causal edge to both the nodes. In our experiments, we used 2 latents per DAG. This results in two new DAGs $\mathcal{D}_1'$ and $\mathcal{D}_2'$. To ensure $\alpha n$ node distance between clusters, we modify $\mathcal{D}_2'$ using random changes until the two MAGs corresponding to the DAGs $\mathcal{D}_1'$ and $\mathcal{D}_2'$ are separated by a distance of $\alpha n$. These two MAGs, denoted by $\mathcal{M}_1^{\text{dom}}$ and $\mathcal{M}_2^{\text{dom}}$ form the dominant MAG for each of the two clusters.

Then, we create $(1-\gamma)M/k = (1-\gamma)M/2$ copies of the dominant MAG and assign it to distinct entities in each cluster. Consider cluster $C_1^\star$ with dominant MAG $\mathcal{M}_1^{\text{dom}}$, and corresponding DAG $\mathcal{D}_1'$. Note that each cluster has $M/k = M/2$ entities. For the remaining entities in $C_1^\star$, we start with $\mathcal{D}_1$ and include 2 latent variables between randomly selected pairs of nodes. Then, we repeat the previous procedure, of performing a series of random insertions or deletions of edges to the DAG until the distance between the corresponding MAG and $\mathcal{M}_1^{\text{dom}}$ increases to $\beta n$. We follow the same procedure for cluster $C_2^\star$ with dominant MAG $\mathcal{M}_2^{\text{dom}}$. Note that in this construction different entities could differ both in latents and their observable graphs. This construction ensures the entities satisfy $(\alpha, \beta)$-clustering property. As an example, see Figure 7 containing two dominant MAGs of the Causal Network *Earthquake*.

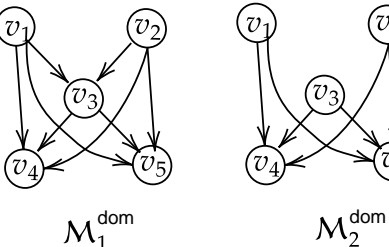

Figure 7: Dominant MAGs of the causal network *Earthquake* constructed using the described procedure.

**Sample Set $S$ Size.** For Algorithm $(\alpha, \beta)$-BOUNDEDDEGREE, we use different sample sizes $S$ ranging from 1 to 3. In Figure 8, we plot the mean value of the maximum number of interventions per entity with change in sample set size.

With increase in sample set size, our Algorithm $(\alpha, \beta)$-BOUNDEDDEGREE requires more interventions (see Lemma C.1) and we observe the same in Figure 8. We chose the smallest size $|S| = 1$ in our experiments, as increasing the size will increase the number of interventions but did not lead to much improved clustering results. As a sample set of size 1 roughly corresponds to around 3 interventions (across all causal networks), we use that for results presented in Table 1.

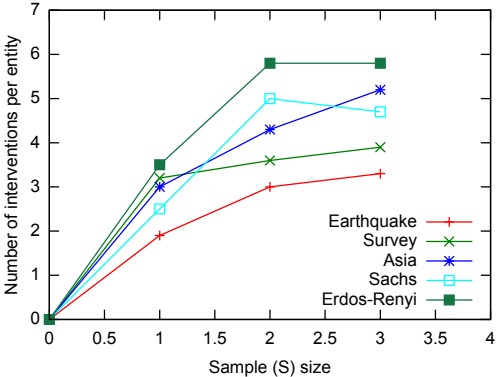
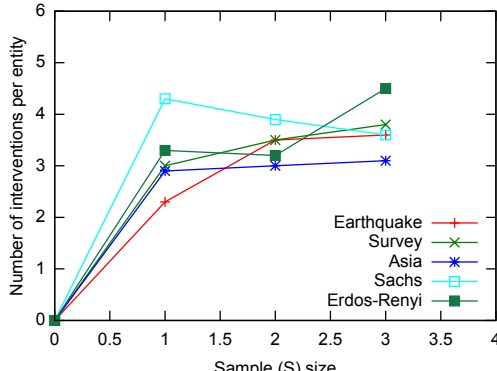

Figure 8: Sample size vs. maximum number of interventions per entity used by Algorithm $(\alpha, \beta)$-BOUNDEDDEGREE.

Figure 9: Sample size vs. maximum number of interventions per entity used by Algorithm $\alpha$-BOUNDEDDEGREE.

| Causal | FCI | | | $\alpha$-BOUNDEDDEGREE (Alg. 6) | | | Maximum |
| Network | Precision | Recall | Accuracy | Precision | Recall | Accuracy | # Interventions |
|---|---|---|---|---|---|---|---|
| *Earthquake* | $0.79 \pm 0.25$ | $0.98 \pm 0.02$ | $0.79 \pm 0.25$ | $1 \pm 0.00$ | $1.0 \pm 0.0$ | $1.00 \pm 0.00$ | 3 |
| *Survey* | $0.79 \pm 0.25$ | $0.98 \pm 0.02$ | $0.79 \pm 0.25$ | $0.89 \pm 0.20$ | $1.0 \pm 0.00$ | $0.89 \pm 0.20$ | 4 |
| *Asia* | $0.84 \pm 0.23$ | $0.98 \pm 0.02$ | $0.84 \pm 0.23$ | $0.89 \pm 0.20$ | $1.0 \pm 0.00$ | $0.89 \pm 0.20$ | 4 |
| *Sachs* | $1.0 \pm 0.00$ | $1.0 \pm 0.00$ | $1.0 \pm 0.00$ | $0.79 \pm 0.25$ | $1.0 \pm 0.00$ | $0.79 \pm 0.25$ | 5 |
| *Erdös-Rényi* | $1.00 \pm 0.00$ | $1.00 \pm 0.00$ | $1.00 \pm 0.00$ | $1.00 \pm 0.00$ | $1.00 \pm 0.00$ | $1.00 \pm 0.00$ | 5 |

Table 2: In this table, we present the precision, recall and accuracy values obtained by Algorithm $\alpha$-BOUNDEDDEGREE and FCI. Each cell includes the mean value along with the standard deviation computed over 10 runs. The last column contains the maximum number of interventions per entity required (including both Algorithm $\alpha$-BOUNDEDDEGREE and the Meta-algorithm) for recovering the DAGs.

**Construction of Clusters from FCI Output.** Our first focus is on recovering the true clustering using Algorithm $(\alpha, \beta)$-BOUNDEDDEGREE. As a baseline, we employ the well-studied FCI algorithm [Spirtes et al., 2000]. We know that FCI returns the Partial Ancestral Graph(PAG) corresponding to the causal MAG using only the observational data. After recovering the PAGs corresponding to the MAGs using FCI, we cluster them by constructing a weighted graph (similar to Algorithm $(\alpha, \beta)$-BOUNDEDDEGREE) defined on the set of entities. For every pair of entities $i, j$, we calculate the number of nodes $n_{ij}$ that share the same neighborhood using the PAGs associated with them, and assign the weight of the edge as $n_{ij}$. This weight captures the similarity between two entities, and whether they belong to the same cluster or not. Now, we use minimum-$k$-cut algorithm to partition the set of entities into $k$ components or clusters. In Algorithm $(\alpha, \beta)$-BOUNDEDDEGREE, we first construct a sample $S$, and perform various interventions based on the set $S$ for every entity to finally obtain the $k$ clusters.

**Setup**. We used a personal Apple Macbook Pro laptop with 16GB RAM and Intel i5 processor for conducting all our experiments. We use the FCI algorithm implemented in [Kalisch et al., 2012]. For every causal network, each experiment took less than 10 minutes to finish all the 10 runs.

### E.2 Learning MAGs under $\alpha$-clustering property

**(Synthetic) Data Generation.** We use following process for each of the five considered causal network (*Asia*, *Earthquake*, *Sachs*, *Survey*, and *Erdős-Renyi*). We construct causal DAGs for $M$ entities distributed among the clusters $C_1^\star, C_2^\star, \cdots C_k^\star$ equally, i.e., $|C_i^\star| = M/k$ for all $i \in [k]$. Again we set $k = 2$. For each of the DAGs $\mathcal{D}_1$ and $\mathcal{D}_2$, we select a certain number of pairs of nodes randomly, and include a latent variable between them, that has a causal edge to both the nodes. In our experiments, we used 2 latents per DAG. This results in two new DAGs $\mathcal{D}_1'$ and $\mathcal{D}_2'$. To ensure $\alpha n$ node distance between clusters, we modify $\mathcal{D}_2'$ using random changes until the two DAGs $\mathcal{D}_1'$ and $\mathcal{D}_2'$ are separated by a distance of $\alpha n$ and are Markov equivalent. Without Markov equivalence,

we observe that FCI always recovers the underlying clusters correctly in the $\alpha$-clustering case.[4] However, existence of Markov equivalent DAGs is a well-known problem in real-world graphs, a popular example to illustrate this comes the "breathing dysfunction" causal graph in Fig. 3 in [Zhang, 2008b]. We create $M/2$ copies of the each of the two DAGs $\mathcal{D}_1'$ and $\mathcal{D}_2'$ and assign it to distinct entities in each of the two clusters.

**Parameters.** We present the following settings for the model parameters, $\alpha$ is at least $0.60$, $M = 40$. For the synthetic data generated using Erdös-Rényi model, we use $n = 10$, probability of edge $p = 0.30$. We ran all of our experiments for 10 times with the stated values and report the results.

**Sample Set $S$ Size.** For Algorithm $\alpha$-BOUNDEDDEGREE, we again tried different set $S$ sizes ranging from 1 to 3. In Figure 9, we plot the mean value of the maximum number of interventions per entity with increase in sample size. It has a same trend as with $(\alpha, \beta)$-clustering (Figure 9). A sample size of 1 roughly corresponds to around 3 interventions, and we use that for the results presented in Table 2.

**Evaluation of Clustering.** We start by results on recovering the clustering using Algorithm $\alpha$-BOUNDEDDEGREE. As a baseline, we again employ the well-studied FCI algorithm [Spirtes et al., 2000]. After recovering the PAGs corresponding to the DAGs using FCI, we cluster them by constructing a similarity graph (similar to the case of $(\alpha, \beta)$-clustering discussed previously) defined on the set of entities. For Algorithm $\alpha$-BOUNDEDDEGREE, we first construct a sample $S$, and perform various interventions based on the set $S$ for every entity to finally obtain the $k$ clusters. We also implemented another baseline algorithm (GREEDY) that uses interventions, based on a greedy idea that selects nodes to set $S$ in Algorithm $\alpha$-BOUNDEDDEGREE by considering nodes in increasing order of their degree in the PAGs returned by FCI. We use this ordering to minimize the number of interventions as we intervene on every node in $S$ and their neighbors. We use the same metrics as the $(\alpha, \beta)$-clustering case.

**Results.** In Table 2, we compare Algorithm $\alpha$-BOUNDEDDEGREE to FCI on the clustering results. For Algorithm $\alpha$-BOUNDEDDEGREE, we use a sample $S$ of size 1, and observe in Figure 9, that this corresponds to about 2 interventions per entity. With increase in sample size, we observed that the results were either comparable or better. We observe that our approach leads to considerably better performance in terms of the accuracy metric with an average difference in mean accuracy of about $0.20$. We observe that entities belonging to the same true cluster are always assigned to the same cluster, resulting in high recall for both Algorithm $\alpha$-BOUNDEDDEGREE and FCI. Further, the higher value of precision for our algorithm is because FCI is unable to correctly detect that there are two clusters, as the DAGs are Markov Equivalent which means that they result in the same PAGs.

Algorithm $\alpha$-BOUNDEDDEGREE outperforms the GREEDY baseline for the same sample ($S$) size. For example, on the *Earthquake* and *Survey* causal networks, Algorithm $\alpha$-BOUNDEDDEGREE obtains the mean accuracy values of $1.0$ and $0.89$ respectively, while GREEDY for the same number of interventions obtained an accuracy of only $0.74$ and $0.64$ respectively. On the remaining causal networks, the accuracy values of GREEDY are almost comparable to our Algorithm $\alpha$-BOUNDEDDEGREE.

After clustering, we recover the DAGs using the Meta-algorithm described in Section D.1, and observe that only one additional intervention is needed. In the last column in Table 2, we report the maximum number of interventions for recovering DAGs, which includes both the interventions used by the Algorithm $\alpha$-BOUNDEDDEGREE and the Meta-algorithm. We observe that our *collaborative* approach uses fewer interventions for MAG recovery compared to the number of nodes in each causal network. For example, in the Erdös-Rényi setup, the number of nodes $n = 10$, whereas we use at most 5 interventions per entity. Thus, compared to the worst-case, cutting the number of interventions for each entity by $50\%$.

---

[4]Again this is not true for $(\alpha, \beta)$-clustering, as shown by our experiments results for that case, because now difference in PAGs between two entities does not automatically imply that those two entities must belong to different clusters.