# OpenReview forum: "Collaborative Causal Discovery with Atomic Interventions"
_NeurIPS.cc/2021/Conference — NeurIPS 2021 Poster_

### Official Review · Reviewer_8cR4 · 2021-07-15

**Rating:** 6
**Confidence:** 3

**Summary:**

The paper introduces an algorithm for learning causal graphs for multiple entities which share some underlying similarities with minimum atomic interventions. The goal is different from the well-studied problem of learning a causal graph from multiple environments in which the data distribution could differ and the aim is to recover a single global graph.

The problem is of learning independent causal graphs is posed from the perspective of finding clusters among the entities where entities in a cluster are closer in some distance metric in comparison to entities in other clusters. This property is termed as $(\alpha,\beta)$ clustering. The task is to minimize the number of interventions needed to recover the causal graphs while distributing the interventions across the clusters to minimize the total intervention cost. A lower bound of $\Omega(\frac{1}{\alpha})$ interventions is shown to be necessary for recovering the clusters as well as the minimum number of atomic interventions needed.

Experiments are performed on multiple datasets with a comparison to the standard FCI algorithm. Results indicate an overall improvement in the metrics for recovering the true causal graph.

**Limitations And Societal Impact:**

There is little direct societal impact and accordingly no extensive discussion about it.

One consideration that can affect the direct application of this in real-world settings would be the high dimensional attributes that can result in dense and complex graphs.
How can the approach be applied in cases where the number of edges in the graph is magnitudes higher than the number of nodes? In such settings, identifying clusters can be computationally challenging.

**Main Review:**

The work is original with respect to introducing clustering as an approach to recovering the similarities in causal graphs across multiple entities. Although not directly related, the task of recovering causal graphs from multiple environments has some similarities with respect to recovering the commonalities across environments. It would be helpful to situate this work in the multi-environment premise and draw out the differences between the two paradigms.

The submission is technically sound and the claims are supported by theoretical and empirical analysis. Although it would be helpful to discuss the specific weaknesses of this work such as the relation with the dimensionality of the nodes, up to what extent does the method succeed when different entities have very few similarities, does it make sense to recover causal graphs for certain entities more accurately than others. For example, in settings such as healthcare where obtaining data from multiple entities (say hospitals) is difficult and hence it may not be possible to recover the causal graphs for different hospitals it would be useful to recover at least one causal graph more accurately to guide clinical decision-making. Similarly, it may not always be possible to collect data about all the nodes in all the entities, in such a case recovering partial causal graphs across entities to potentially recover a global graph is also an interesting direction.

The paper is well-organized. It is generally clear to read.

The approach can definitely guide further development in the field of causal discovery and opens up several interesting directions.


**Time Spent Reviewing:**

9

---

> ### Author Response · Authors · 2021-08-10
> **Authors' response**
>
> > 1. “Multi-environment premise..”:
>
> Thanks for this suggestion. We do not see a direct connection between the two settings, but would be happy to explore if the reviewer had something in mind with respect to our problem setting.
>
>
> > 2. “... such as the relation with the dimensionality of the nodes, up to what extent does the method succeed when different entities have very few similarities”:
>
> One of the key contributions is that our algorithms require only a few interventions per entity, all our bounds are in terms of the number of nodes ($n$) in the system. In fact, our bounds depend only on logarithmic in the number of nodes in degree-bounded graphs. We capture similarities between entities in the clusters using both inter-cluster ($\beta$) and intra-cluster ($\alpha$) distances. As Theorem 3.4 suggests a good case for the bound is when $\alpha-\beta$ is high (close to $1$), and vice-versa a bad case is when $\alpha-\beta$ is low (close to $0$). We would be happy to add these points.
>
>
> > 3. “does it make sense to recover causal graphs for certain entities more accurately than others. For example, in settings such as healthcare where obtaining data from multiple entities (say hospitals) is difficult and hence it may not be possible to recover the causal graphs for different hospitals it would be useful to recover at least one causal graph more accurately to guide clinical decision-making.”:
>
> We completely agree with the reviewer that in some scenarios it might be a valid goal to recover certain graphs more accurately than others. We believe that our introduced framework of collaboratively learning causal graphs could be adapted to frame these questions.
>
>
> > 4. “Similarly, it may not always be possible to collect data about all the nodes in all the entities, in such a case recovering partial causal graphs across entities to potentially recover a global graph is also an interesting direction”:
>
> We want to emphasize that we don’t need interventional  (experimental) data from all the nodes for all entities. Instead, we spread the interventions across entities with each entity performing interventions on very few nodes, and for each entity, we only need data to support the local CI tests. We agree that recovering partial causal graphs is an interesting future direction to consider in settings where interventions on certain variables are not possible for most entities.
>
>
> > 5. “.. where the number of edges in the graph is magnitudes higher than the number of nodes..”:
>
> Our results when MAGs satisfy the $\alpha$-clustering property are independent of the degree of the MAG. So, for identifying clusters, our algorithms do not depend on the number of edges in the MAG. For learning causal graphs that satisfy the $(\alpha, \beta)$-clustering property, the number of interventions per entity used by our algorithms depends linearly on the maximum degree of the graph. Such dependencies are not uncommon for learning causal graphs (Kocaoglu et. al. 2017, ‘Experimental Design for Learning Causal Graphs with Latent Variables’). This dependence on maximum-degree is justified as most real-world causal bayesian networks are known to be sparse (see e.g., https://www.ccd.pitt.edu/wiki/index.php/Data_Repository).

---

> > ### Comment · Reviewer_8cR4 · 2021-08-30
> > **Reponse to the authors of Paper6478**
> >
> > I would like to thank the authors for their clarifications about all the issues raised. I appreciate the author's comments about the multi-environment setting. There seems some potential to extend the framework to a multi-environment setup as given in https://proceedings.neurips.cc/paper/2017/file/62889e73828c756c961c5a6d6c01a463-Paper.pdf, although this is an interesting future direction and does not affect my score. Thus, I keep my score the same and thank the authors for their feedback.

---

### Official Review · Reviewer_Ec77 · 2021-07-17

**Rating:** 6
**Confidence:** 4

**Summary:**

The paper proposes an approach to learn multiple causal graphs, in the presence of latents, collaboratively and simultaneously by means of clustering and by performing atomic interventions.
First, the work proposes the property $(\alpha,\beta)$-clustering where graphs in different clusters are "different" by a factor of $\alpha$ at least, while graphs within the same cluster are "different" by a factor of $\beta$ at most. Using this property, an approximate algorithm is proposed to (1) learn the clusters such that the $(\alpha,\beta)$-clustering property is satisfied, and (2) learn one dominant graph in each cluster assuming the clusters are more homogeneous.
Next, a special case of the clustering property is considered where all the graphs within the same cluster are identical, i.e. $\beta=0$, and a learning algorithm is proposed with a lower bound on the number of interventions required to learn the graphs than the general case.
Finally, the proposed algorithms are evaluated using data generated from synthetic and real causal graphs.

**Limitations And Societal Impact:**

The authors address the limitations and potential negative societal impact of their work adequately.

**Main Review:**

**Originality:**
The central idea of the paper on collaborative causal discovery using atomic interventions is somewhat original, some of the techniques such as learning a node's neighbours (Algorithms 1 & 2) are based on well-known techniques, and the related work is adequately cited.

**Quality:**
My evaluation is based on the main body of the paper only and not the supplementary material. That being said, the work seems to be technically sound. However, a couple of issues are noted below:

+ Using MAGs: Modeling the causal graphs using MAGs is a major weakness of the work when considering interventions. MAGs are attractive under Markov equivalence [27,28] and under soft interventions [10,14] because the true causal diagram (DAG with latents or ADMG) is not differentiable from its equivalent MAG. This is not true under hard interventions as considered in this work.
For instance, consider the causal graph $G=\\{ A\rightarrow B\rightarrow C, B\leftarrow L \rightarrow C \\}$ and the corresponding MAG $M=\\{ A\rightarrow B\rightarrow C, A\rightarrow C \\}$. Both graphs are Markov equivalent, yet they are differentiable under $do(B)$.
The challenge discussed in lines 131-133 is a limitation in atomic interventions rather than a justification for adopting MAGs.

+ $(\alpha,\beta)$-clustering: The parameters $\alpha,\beta$ are given to Algorithm 3 as input and it is probably not possible to obtain a valid clustering of the graphs for any pair of parameters. If so, then finding the right/convenient parameters for clustering is a search task by itself. This issue is reflected by the antecedent of Lemma 3.3. I appreciate your thoughts on this.

+ Proposition 3.2: This lower bound is computed assuming that the PAG does not exist and we are learning the MAG purely from a sequence of atomic interventions. However, the algorithm "RECOVERG" clearly utilizes the PAG through Algorithms 1 & 2. I would assume that the bound could be improved in the presence of observational data represented by the PAG.

**Clarity:**
The problem is well motivated in the introduction with real-world examples. Furthermore, the problem statement is clearly formulated and illustrated in Section 2. The rest of the paper could benefit from some examples to illustrate the algorithms and the key ideas, especially in Section 4 which feels somewhat dull (namely, "Overview of Algorithm $\alpha$-GENERAL"). Also, the merging step in "Overview of Algorithm $(\alpha,\beta)$-RECOVERY" is ambiguous and further elaboration could be helpful.

+ Minor typos:

Line 399, "scenarios where are multiple" --> where *there* are multiple.

Line 170, "With right setting of" --> with *a/the* right setting

**Significance:**
The task of causal discovery is important to various fields and the proposed idea of collaborative learning through clustering and efficient interventions can be relevant to a wide audience. However, the work is limited by the issues raised in the "Quality" section.

**Time Spent Reviewing:**

5-6

---

> ### Author Response · Authors · 2021-08-10
> **Authors' response**
>
> > 1. “Using MAGs: ....”:
>
> When not all variables of interest can be measured, DAGs between these observed variables are not sufficient to represent the observed distribution, since latent variables may introduce confounding effects between the observed variables. In these cases, it is very common to model the observed variables through MAGs and it comes with many desirable properties (see e.g., Richardson et al. 2002, ‘Ancestral Graph Markov Models’).
>
> The example the reviewer points out is correct. Using single vertex (atomic) hard interventions, we can differentiate MAGs from DAGs for some specific graphs. However, it is not generally true for any pair of MAG and a DAG. Consider the DAG $G=$ { $A \rightarrow B \rightarrow C, A \leftarrow L1 \rightarrow C, A \leftarrow L2 \rightarrow B, B \leftarrow L3 \rightarrow C $} with latents $L1$, $L2,$ and $L3$; and a MAG $M=$ { $A \rightarrow B \rightarrow C, A \rightarrow C $}. We can observe that both these causal graphs are Markov equivalent. However, unlike the example mentioned by the reviewer, we cannot distinguish these two graphs using any single vertex interventions. Such examples can be constructed for distinguishing two DAGs as well (Fig. 4).
>
>
>  As described in our Introduction (Lines 21-26), for practical reasons, our choice of interventions is restricted to atomic interventions (non-atomic interventions are just too hard to implement in practice). However, we note that our choice of using MAGs is not based on the choice of the interventional setup (atomic or not), but rather the fact that the MAGs are an attractive way to model the causal structure in presence of latents. Our setup does benefit from the fact the learning of MAGs is compatible with atomic interventions (Lines 130-135).
> We will revise this discussion in the Introduction to clarify this point.
>
>
> > 2. “..parameters of $(\alpha, \beta)$-clustering..”:
>
> We discuss the setting of the parameters in our submission (Lines 193-198), and also summarized here for sake of convenience. Firstly for all our algorithms, a lower bound for $\alpha$ and upper bound for $\beta$ is sufficient. In practice, a clustering of the PAGs (generated from FCI algorithm) can provide guidance about these bounds on $\alpha,\beta$, or if we have additional knowledge that $\alpha \in [1-\epsilon, 1]$ and $\beta \in [0, \epsilon]$ for some constant $\epsilon > 0$, then, we can use a binary search based guessing strategy for $(\alpha,\beta)$ starting from $\alpha = 1-\epsilon, \beta = \epsilon$. We divide the interval for $\alpha$, given by $[1-\epsilon, 1]$, into $\epsilon n$ values separated by $1/n$, and binary search among these values. Similarly, we search for $\beta \in [0, \epsilon]$. As we search for both $\alpha, \beta$ simultaneously, this increases our intervention bounds (number of interventions per entity) by a multiplicative factor of ${\log^2 (\epsilon n)}/{(1-2\epsilon)^2}$.
>
>
> > 3. “Regarding Proposition 3.2”:
>
> Note that our lower bound for the number of interventions per entity is for the first step of identifying the underlying clustering. Similar to our upper bounds, our lower bound considers the worst-case. Consider one such worst-case scenario where the MAGs satisfy the $\alpha$-clustering property and are all Markov equivalent. This implies that the PAGs obtained using FCI are identical and will not be helpful in identifying the clusters. We will add this clarification to our submission. In practice, we agree with the reviewer, that the information available in the PAGs could be useful to reduce the number of interventions.
>
>
> > 4. "Elaboration of Algorithm $(\alpha,\beta)$-$\text{Recovery}$”:
>
>  We thank the reviewer for their suggestion, and we will include examples to illustrate the main ideas in Section 4.
>
> After recovering the clustering using Algorithm $(\alpha, \beta)$-$\text{BoundedDegree}$, our goal is to learn the causal graphs. Using Algorithm $(\alpha,\beta)$-$\text{Recovery}$, we show that we can learn these graphs approximately up to a distance approximation of $\beta n$.
>
>  In a cluster $C^{\star}_a$, we construct a partitioning of MAGs such that two MAGs belong to a partition if they are equal. The MAG corresponding to the largest partition is called dominant MAG. Using our algorithm, we learn the dominant MAG correctly and return it as an output. As all the MAGs in the cluster $C^{\star}_a$ satisfy $(\alpha, \beta)$-clustering property, the dominant MAG is within a distance of $\beta n$ from the true MAG and therefore is a good approximation of the true MAG.
>
> For learning dominant MAG, there are two steps. First, we select a node uniformly at random for every entity and intervene on the node and its neighbors to learn all the edges incident on the node. Next, we construct the dominant MAG by combining the neighborhoods of each individual node. Let $u$ be any node and $T_u$ denote the set of all entities which intervened on $u$ in the first step. Now, among all the neighborhoods identified by the entities in $T_u$, we do not know which of them correspond to that of the dominant MAG. In order to solve this issue, we use a threshold-based approach and assign a score to every entity in $T_u$. The score of an entity $i$ is the number of entities in $T_u$ that has the same neighborhood of $u$ as that of $i$. Finally, we select the entity with the maximum score and assign the neighborhood of the entity as the neighborhood of $u$ for the dominant MAG (Lines 12-15 in Algorithm $(\alpha,\beta)$-$\text{Recovery})$.
>
>  We argue that if the cluster size is large (see Theorem 3.4), the neighborhoods of nodes using entities with maximum scores are equal to that of the dominant MAG. This is because the dominant MAG has the largest partition size, and if a sufficiently large number of entities (across all partitions) are assigned node $u$, then, many of them will be entities from the dominant MAG partition. So, the scores of entities in the dominant MAG partition will be higher compared to the scores of entities from other partitions.

---

### Official Review · Reviewer_8Ccq · 2021-07-18

**Rating:** 6
**Confidence:** 3

**Summary:**

The main goal of the paper is to perform Collaborative Causal Discovery. The main contributions of the paper are the following
* Define a new framework for collaborative discovery. The authors consider the case of several entities, each being associated to a DAG containing both observed and latent variables. The idea is to recover the so-called Maximal Ancestral Graph of each entities under a clustering assumption, stating that the MAG are more or less similar inside a cluster, and different when considering different clusters clusters
* Under this clustering assumption, is provided an algorithm allowing to recover the MAG of each entity with a minimal number of interventions. The algorithm is a two step procedure, performing first a clustering of the entities and thereafter estimating the MAG of each entity
* Numerical experiments complete the paper
* In Appendix,  details about MAG are given as well as proofs on the minimality of the numbers of interventions to recover the causal graphs


**Limitations And Societal Impact:**

Everything is OK from this point of view

**Main Review:**

* The problem addressed in this paper is relevant. Two scenarios where collaborative causal discovery can be performed are given that makes sense. I wonder how general this setting is and in particular if it is totally new or if it encompass already studied framework
* The idea of considering MAG instead of DAG is well justified and is smart. Details are given on MAG in Appendix A.
* I wonder if the notion of atomic intervention is the same for a MAG than a DAG.  Is there any do calculus in this setting? How in practice can we intervene on a MAG? Can you give any references? The reference given for MAG is a relevant one and explains quite well why MAG are interesting but details about interventions in this setting could help
* Numerical experiments are performed on classical benchmark with no more than a dozen of nodes. How could be extended the approach when we have more variables? How can we decide of the number of latent variables when dealing with practical data
* You only compare with FCI. I understand that you want to compare with algorithm dealing only with observational data but you mention also  other algorithm trying to minimize the number of interventions. Even if your setting is different, on the examples studied in Section 5, could some comparison be done?

**Time Spent Reviewing:**

3

---

> ### Author Response · Authors · 2021-08-10
> **Authors' response**
>
> > 1. “Generality of our collaborative causal learning framework”:
>
> While the notion of collaborative learning is not new for machine learning, modeling the problem of causal discovery through the lens of collaborative learning is new. In general, causal discovery, in presence of latents, is a hard problem because of its reliance on interventions. Our collaborative learning framework provides a clear way of reducing the number of interventions per entity. We believe that this novel framework will inspire further work in this area.
>
>
> > 2. “... Atomic Interventions for MAG ...”:
>
> The interventions we use in our paper are defined with respect to the underlying DAG of a given MAG (note that the mapping from DAG to MAG is unique). In our model, (Lines 121-122) we assume that there are n DAGs  $\mathcal{D}_1,\dots,\mathcal{D}_M$ one for each entity in $[M]$, with $\mathcal{M}_1,\dots,\mathcal{M}_M$ being the corresponding MAGs. The interventional distributions used in our paper are defined with respect to these DAGs, therefore the do-calculus defined for DAGs is applicable in our setting as well. However, the inference of the causal structure happens on the MAGs. We want to highlight that even though we do not know the structure of the underlying DAG (the graph on observable nodes as well as the latents), we are able to learn the structure of the MAG, using atomic interventions (defined over the DAG). Also answering the reviewer question, while not directly relevant for us, there is a well-defined do-calculus with respect to MAGs (see e.g., Zhang 2007, ‘Generalized Do-Calculus with Testable Causal Assumptions’).
>
> >3. “Larger Scale Numerical Experiments.”:
>
> We are happy to present results on larger node set graphs. Below we present the results on a 50 node Erdos-Renyi random graph setting when they satisfy the $(\alpha,\beta)$-clustering property. We use the same setup for these experiments as those presented in our submission (Section 5), and report the results averaged over multiple runs (the variance across runs were negligible and omitted here). We observe that our algorithm outperforms FCI with similar margins as described in our submission.
>
> $n=50$ Results: In the case of a 50 node random graph, with probability of edge being present equaling 0.04, the average clustering accuracy and F-score values of FCI were $0.48$ and $0.64$ respectively. Meanwhile, our algorithm recovered the true clusters exactly (i.e., accuracy and F-score values of $1.0$ and $1.0$). Our algorithm uses only at most 8 interventions per entity.
>
> | $n=50$      | Accuracy | F-score     |
> | :---        |    :----:   |          ---: |
> | FCI     | $0.48$       |  $0.64$   |
> | Our Algorithm (Alg 3)   | $1.0$       | $1.0$      |
>
> We would be happy to add these results to the paper and perform more experiments if the reviewer recommends.
>
> > 4. “How can we decide on the number of latent variables when dealing with practical data”:
>
> We do not have to decide on the number of latents. As MAGs encode the latent variables using bi-directed edges, MAGs are well-defined independent of the number of latents. Therefore, we do not need this knowledge even in our algorithms. In our experiments, as we use the DAGs from the real-world causal networks containing only observable nodes, the number of latents and the location of the latents are design choices that we make only for evaluation purposes.
>
>
> > 5. “..on comparison with FCI only.. ":
>
> Note that in addition to FCI, we also compare to another baseline called GREEDY (Lines 366-370) that uses interventions. We compare our results with GREEDY in Lines 386-390 of our submission. We compare against both FCI and GREEDY also in the $\alpha$-clustering case, results in Appendix E.2.

---

### Official Review · Reviewer_H6LC · 2021-07-20

**Rating:** 6
**Confidence:** 2

**Summary:**

This paper proposes a method for causal discovery by aggregating data from different data sources into clusters. Those clusters are similar with respect to the underlying graphical structure that generates the data. Based on these clusters, the authors an efficient algorithm in terms of number of interventions required to estimate the underlying graph.

**Limitations And Societal Impact:**

None.

**Main Review:**

I enjoyed reading this technical paper that has also a convincing experimental evaluation. I am not familiar with the field so my comments here are based on my limited understanding of the different presented notions. The paper is well-written, although I would find more useful a better and more thorough literature review to understand how this paper relates to other works in the field. The article is well written, although quite dense in notation which requires going back and forth between the main text and the appendix.

I have a couple of questions:

First, the input to the $(\alpha, \beta)$-BoundedDegree algorithm contains the PAGs given by FCI. The results in Lemma 3.3 and Thm 3.4 are statements with high probability but with respect to the distribution introduced by the algorithm. What happens if the initial PAGs are incorrectly estimated?

More generally, I am surprised that there is no analysis of the initial graph estimation procedure that is based on conditional independence tests. Isn't the large number of simultaneous tests an issue? Your clustering algorithm is built on objects that are estimated and not ground truth or actual data samples, don't you take into account that uncertainty at all?

The $(\alpha, \beta)$-Recovery algorithm gives as output the dominant MAG for each entity, which Thm3.4 shows is close enough with high-probability to the true MAG. Does that imply that, even with an infinite number of atomic interventions, it is impossible to separate two MAGs from the same initial cluster?

**Time Spent Reviewing:**

2.5

---

> ### Author Response · Authors · 2021-08-10
> **Authors' response**
>
> > 1. “On PAG estimation”:
>
> Yes, the reviewer is correct that the current writeup assumes (in Algorithms 3 and 6) that the initial PAGs are estimated correctly.
>
> (a) We rely on this assumption because PAG estimation is a very well-studied problem in causal discovery from both a theoretical and practical perspective. We mention the FCI (Fast Causal Inference) procedure [26] whose soundness and completeness are well understood (see e.g., Spirtes et al. 1999, ‘An algorithm for causal inference in the presence of latent variables and selection bias’, and Zhang 2008, ‘On the completeness of orientation rules for causal discovery in the presence of latent confounders and selection bias’). Also, recent variations of FCI such as Really Fast Causal Inference (RFCI) have sped up the FCI procedure (Colombo et al. 2012, ‘Learning high-dimensional directed acyclic graphs with latent and selection variables.’​​). Today FCI/RFCI  procedures are commonly used in practice, with various implementations available [12].
>
> (b) Note, for all our algorithms and bounds, all that we require from the PAGs is that they have the correct (undirected) skeleton as their corresponding MAGs, i.e., we could just ignore all the directed edges in the initial PAGs and replace them with $\circ-\circ$ edges before using them in our algorithms,  and this would not change our results.
>
> (c) Finally, we could even relax our assumptions and tolerate error even in skeleton estimation, a point that is currently not noted in the paper. The idea is simple, and we sketch it here. Consider for example the $\alpha$-clustering case, i.e., the MAGs $\mathcal{M}_1,\dots,\mathcal{M}_M$ satisfy the $\alpha$-clustering assumption with true clusters $C^\star_1,\dots,C^\star_k$. Now consider the setting where we have errors in the PAG skeleton estimation. Let $\mathcal{U}_1,\dots,\mathcal{U}_M$ be the true skeletons of the MAGs $\mathcal{M}_1,\dots,\mathcal{M}_M$. Consider for each MAG $\mathcal{M}_i$ a corrupted counterpart $\mathcal{M}^\text{corr}_i$ with the guarantee that $d(\mathcal{M}_i,\mathcal{M}^\text{corr}_i) \leq (\beta/2) n$. These corrupted MAGs are only constructed for the sake of proof, and are not actually present. Assume that the skeleton estimation is not precise and instead of $\mathcal{U}_1,\dots,\mathcal{U}_M$, it produces the skeletons $\mathcal{U}^\text{corr}_1,\dots,\mathcal{U}^\text{corr}_M$, associated with these corrupted MAGs $\mathcal{M}^\text{corr}_1,\dots,\mathcal{M}^\text{corr}_M$. By triangle inequality, it is easy to observe that the MAGs $\mathcal{M}^\text{corr}_1,\dots,\mathcal{M}^\text{corr}_M$ satisfy $(\alpha-\beta,\beta)$-clustering assumption. If $\beta < \alpha/2$, then using Algorithm 3 on $\mathcal{U}^\text{corr}_1,\dots,\mathcal{U}^\text{corr}_M$ with parameter $\alpha$ replaced by $\alpha-\beta$ will guarantee that we recover the true clusters $C^\star_1,\dots,C^\star_k$. This follows because any pair of entities $i,j$ that were originally in the same true cluster will still remain together in the same cluster, even under corruption, as their corrupted MAGs will be at most $\beta n < (\alpha/2) n$ distance apart. Similarly, if $i,j$ belonged to different true clusters then they will still remain in different clusters, even under corruption,  as their corrupted MAGs will be $> (\alpha/2) n$ distance apart.  Also, if the corrupted MAGs satisfy the conditions in Theorem 3.4, we can recover the dominant MAG. With the right set of parameters, this argument can also be extended starting from an $(\alpha,\beta)$-clustering.  We believe that this in fact shows the versatility of our framework, and if the reviewer recommends, we would be happy to add these details formally in the paper.
>
> > 2. “Uncertainty in CI Tests”:
>
> The reviewer is correct in these assertions. However, we omitted these details in part because of the previous work on this topic. Since the convergence rates of CI tests are well-known (e.g., Neykov et al. 2020, ‘Minimax Optimal Conditional Independence Testing’), one could recover the sample size bound with any of these PAG estimation procedures for the desired Type 1 error bound. We would be happy to add a discussion on this topic. Note that in our experiments, we do run CI tests on actual data samples generated by our model.
>
>
> > 3. “....Does that imply that, even with an infinite number of atomic interventions, it is impossible to separate two MAGs from the same initial cluster?”:
>
> No, as mentioned in Lines 206-207 (also Lemma B.4 in Appendix B), $n$ atomic interventions suffice to recover the MAG on $n$ variables correctly. Therefore, if we are willing to perform $n$ interventions on each entity, we can recover the exact MAGs correctly and not just the dominant MAG of each cluster.

---

### Author Response · Authors · 2021-08-10
**Authors' response**

We thank all the reviewers for their helpful reviews. We appreciate the consensus over the relevance and originality of our idea. We have addressed each of the reviews separately and would be happy to answer additional questions that the reviewers may have.

---

### Decision · Program_Chairs · 2021-09-27

**Decision:**

Accept (Poster)

**Comment:**

All reviewers are favorable after author responses and discussion.

The paper considers collaboratively learning different MAGs which have a clustering property. Authors show an efficient way to learn all the MAGs, with the smallest number of atomic interventions per entity. It scales in the degree of the MAG and logarithmic in the number of MAGs.

The collaborative setting is novel, the interventional complexity under the clustering assumption is very interesting. This significantly expands the score of learning through intervention of related causal models that could be related but different (apart from being just intervened versions of each other).

Note: Reviewer 8Ccq indicated that she/he would raise the score to 7 but did not follow through after the discussion period.